

# Towards Retrieving Cloud Top Entrainment Velocities from MISR Cloud Motion Vectors

Arka Mitra[1], Virendra P. Ghate[1]

[1]Environmental Sciences Division, Argonne National Laboratory, Lemont, IL, USA 60439

*Correspondence to*: Virendra P. Ghate (vghate@anl.gov)

**Abstract.** Although important, direct retrievals of entrainment rates in cloud-topped planetary boundary layer (PBL) remain elusive. Here we present a novel technique for retrieving cloud-top entrainment velocities using only Multi-angle Imaging Spectro-Radiometer (MISR) stereoscopic retrievals of cloud-motion vectors (CMVs) and cloud-top heights (CTHs). Mesoscale vertical air velocity at CTH is diagnosed from the continuity equation and then used to derive entrainment velocities

from the PBL mass-budget equation. The uncertainties in the utilized CTHs and CMVs are propagated to derive systematic and random retrieval uncertainties. The algorithm is demonstrated through a case of marine stratocumulus deck off the California coast, with comparisons made against the output from European Center for Medium-range Weather Forecasting (ECMWF) reanalysis model (ERA5). MISR low-cloud CTH for this case were lower than the ERA5 reported PBL depth by $189 \pm 87$ m. These differences in cloud top heights partly modulate the differences in the ERA5 and MISR horizontal winds,

with larger differences in meridional over zonal wind components. Average difference between ERA5 and MISR derived mesoscale vertical air motion at cloud top was $0.14\pm0.73$ cm s$^{-1}$, while the same for entrainment rate was $-0.09\pm0.46$ cm s$^{-1}$. Fractional uncertainty is lower than 25% when the retrieved mesoscale vertical air motion is stronger than $\pm0.04$ cm s$^{-1}$ and entrainment velocities are stronger than 0.03 cm s$^{-1}$. These results showcase the ability to derive mesoscale vertical air motion and entrainment rates from MISR observations and motivate its extension to a generate a global climatology leveraging its full

23-year record (2000-2022).



## 1 Introduction

Boundary layer cumulus and stratocumulus clouds are routinely observed over land and oceans from the tropics to the poles.

As these clouds reflect more solar radiation to space compared to the underlying surface, they exert a strong radiative influence on the Earth's atmosphere (Wood, 2012; Klein and Hartmann, 1993; Bony and Dufresne, 2005). These low clouds are intimately coupled to the turbulence in the planetary boundary layer (PBL) that is modulated by surface fluxes, cloud top radiative cooling, wind shear and cloud top entrainment mixing (Bretherton et al., 2007; Moeng et al., 2005; Mellado et al., 2014; Wood, 2012; de Lozar and Mellado, 2017; Stevens, 2005). The mixing of the warm and dry air from above the cloud

top into the cloud layer termed as entrainment, plays a crucial role in modulating cloud microphysical and radiative properties, thereby controlling cloud lifetime (Ackerman et al., 2004; Bretherton et al., 2007; Stevens, 2005; Wood, 2012; Eastman and Wood, 2018). The effects of entrainment mixing on PBL and cloud properties have been vastly debated and remain challenging to be accurately represent in Earth System Models (ESM) and operational weather models (Nam et al., 2012; Vial et al., 2013). Recent studies have also shown entrainment mixing to modulate the responses of cloud and rain properties to changes in

background aerosol concentrations (Grosvenor et al., 2018; Xu et al., 2022; Luo et al., 2020). Despite its importance, largely due to the difficulty in measuring, there have been few observations of entrainment rates, especially over remote oceanic regions.

As entrainment happens at very fine spatial (less than 10 m) and temporal (1-10 second) scales, the entrainment rates need to be derived from high resolution measurements (de Lozar and Mellado, 2017; Mellado, 2017). Observations of conserved

variables and vertical air motion from a level leg at the cloud top have been used to derive entrainment velocities (Faloona et al., 2005; Gerber et al., 2013; Malinowski et al., 2013). However, these estimates suffer from sampling bias, are sporadic, and are cost intensive. In addition, they can be only made from slow moving airborne platforms like the Twin Otter or unmanned aerial systems (UAS). Entrainment rates can also be retrieved from estimates of turbulence kinetic energy (TKE) derived from observations made by cloud radars (Albrecht et al. 2016; Shupe et al., 2012; Borque et al., 2016). However, these estimates

can only be made for non-precipitating clouds, as the presence of precipitation inhibits retrievals of TKE dissipation rates. Lastly, estimates of entrainment rates can also be derived through boundary layer mass conservation framework (Caldwell et al. 2005; Ghate et al. 2019). Although this approach yields dynamically consistent entrainment rates, the observations are very hard to make as it requires data from multiple radiosondes, and a separate estimate of mesoscale vertical air motion.

Entrainment rates have also been derived from satellite observations. Painemal and Zuidema (2011) retrieved entrainment rates by employing the mass-budget equation to the cloud top height estimates from the polar orbiting and geostationary satellites. Although novel, this approach requires utilizing mesoscale vertical air motion estimates from reanalysis models. In addition, due to the inherent uncertainty of retrieving cloud top heights from the measured cloud top temperatures, it is challenging to assess the true uncertainty of the derived entrainment rates (Salonen et al., 2015; Marchant et al., 2020; Minnis




et al., 2020). More recent satellite-based global retrievals and parameterizations of of entrainment rates (Zhu et al., 2025 a, b) have also exhibited the same reliance of reanalysis inputs for characterizing key boundary layer properties.

The Multi-angle Imaging SpectroRadiometer (MISR) aboard NASA's Terra satellite offers a unique opportunity to address these challenges. MISR simultaneously retrieves cloud-top heights (CTHs) and horizontal cloud-motion vectors (CMVs) from stereoscopic imagery at nine viewing angles. Unlike many other satellite techniques, these retrievals do not rely on ancillary thermodynamic profiles such as temperature soundings. For low clouds with tops below 3 km, CTH uncertainties are typically ~0.3 km, while CMV biases are ~0.5 m s$^{-1}$ with random errors of 2–3 m s$^{-1}$ (Horváth, 2013; Marchand et al., 2007; Mueller et al., 2017; Mitra et al., 2021). Because stratocumulus cloud tops generally coincide with the PBL inversion, MISR provides a direct view of PBL-top winds. Moreover, the joint retrieval of heights and winds enables estimation of horizontal wind divergence at cloud top, avoiding the larger height-assignment errors that affect geostationary AMVs (Velden and Bedka, 2009; Salonen et al., 2015; Córdoba et al., 2017). With more than two decades of continuous observations (2000–2022), MISR offers a uniquely homogeneous dataset for entrainment studies made consistently at very similar retrieval times (10:30 am local time).

Here we present a new methodology to retrieve mesoscale vertical air velocity ($w$) at the cloud top and the associated entrainment velocity ($w_e$). The approach applies continuity and mass-budget framework to MISR stereo winds and cloud top heights. Retrieval uncertainties are quantified based on the well-characterized errors in MISR wind vectors and cloud-top heights. The technique is demonstrated through a case study of a persistent summertime stratocumulus deck off the US West Coast (4th June 2018). The MISR-derived entrainment rates are compared to those from ERA5 reanalysis. Beyond this case study, the retrieval method can be extended to the global, multi-decadal MISR record to generate a long-term observational climatology of cloud-top entrainment rates. Such a dataset would provide a benchmark for evaluating entrainment parameterizations in range of atmospheric models and studying regional and seasonal variability of low-cloud entrainment.

In Section 2, we first describe the mathematical framework for deriving entrainment velocity and its uncertainty, followed by the datasets. In Section 3, the technique is applied to data from a marine stratocumulus case, and the results are compared against ERA5 reanalysis. The key findings are summarized in the last section together with discussion on the broader implications of a global MISR-based entrainment dataset.

## 2 Retrieval Technique and Data

The mesoscale vertical air motion and cloud top entrainment rates are kinematically related to each other, and hence it is useful to retrieve both simultaneously. The novelty of our proposed technique is the simultaneous retrieval of mesoscale vertical air motion, and entrainment rates from satellite-based (MISR) observations alone (i.e., without leveraging any "ancillary" data). Data handling is intentionally minimal: subset to the overpass window and domain, quality filters, and co-registration to



common grids used for gradient estimation and diagnostics. Below we describe the theory behind the retrieval algorithm, followed by uncertainty estimation for individual retrievals and then the datasets used for the study.

**2.1 Mathematical Formulation & Algorithm Implementation**

To solve for the entrainment rate ($w_e$) and mesoscale vertical velocity ($w$) using the knowns - cloud-top height (CTH, or $H_T$) and horizontal wind velocity at CTH ($u, v$) - we need to formulate two equations: a continuity equation and a mass-budget equation for the cloud-topped PBL.

The continuity equation for an incompressible atmosphere in the PBL relates the horizontal divergence of the wind field to the vertical gradient of vertical wind velocity. For a layer extending from the surface ($z = 0$) to the cloud-top height ($z = H_T$), the continuity equation is:

$$\frac{\partial u}{\partial x} + \frac{\partial v}{\partial y} + \frac{\partial w}{\partial z} = 0 \tag{1}$$

Above equation can be integrated from the surface ($z=0$) to the cloud top ($z = H_T$).

$$\int_0^{H_T} \left[\frac{\partial u}{\partial x} + \frac{\partial v}{\partial y}\right] dz + [w(H_T) - w(0)] = 0 \tag{2}$$

Assuming that the vertical air motion at the surface is zero (Wood and Bretherton, 2006) and that the changes in the profiles of horizontal winds (first 2 terms of Equation 1) are constant over the PBL depth yields:

$$H_T \left[\frac{\partial u}{\partial x} + \frac{\partial v}{\partial y}\right] + w = 0 \tag{3}$$

Thus, solving the continuity equation results in an observational estimate of the large-scale velocity as:

$$w = -H_T \left[\frac{\partial u}{\partial x} + \frac{\partial v}{\partial y}\right] \tag{4}$$

The RHS of the above equation can be estimated from spatial gradients of the horizontal wind components at the cloud top ($u, v$) and cloud top height ($H_T$) provided by MISR CMVs. The assumptions made in deriving $w$ are evaluated in the Appendix section.



The entrainment velocity $w_e$ represents the velocity at which free tropospheric air enters the PBL across the cloud top. Hence, for a well-mixed boundary layer, the local change and advection of boundary layer depth is balanced by the sum of mesoscale vertical air motion and entrainment velocity.

$$\frac{\partial H_T}{\partial t} + u\frac{\partial H_T}{\partial x} + v\frac{\partial H_T}{\partial y} = w + w_e \tag{5}$$

Since we are dealing with satellite snapshots, by assuming steady state the local change in cloud top height (i.e., the first term in Equation 5) goes to zero. Hence, the entrainment velocity can be derived using the following equation:

$$w_e = u\frac{\partial H_T}{\partial x} + v\frac{\partial H_T}{\partial y} - w \tag{6}$$

Thus, by employing equations (4) and (6) to MISR CMV data (wind vectors and CTHs), the mesoscale vertical air motion at the cloud top and entrainment rates can be simultaneously derived.

*Uncertainty Analysis:* We quantify the systematic uncertainty (accuracy) and the random uncertainty (precision) for the retrieved cloud-top vertical air velocity ($w$) and entrainment velocity ($w_e$) from the measurement uncertainties in the MISR observations used to solve Equations (4) and (6). To derive the analytical forms of the uncertainty estimates over 2D fields of cloud top heights, $z = H_T(x, y)$ and horizontal wind velocity, $\vec{V}(u, v)$, the continuity estimate from Eq. (4) is reframed as

$$w = -H_T . D, \qquad with\ D = \frac{\partial u}{\partial x} + \frac{\partial v}{\partial y} \tag{7}$$

The entrainment estimate from Eq. (6) is reframed as

$$w_e = A - w, \qquad with\ A = \vec{V}.\nabla H_T = u\frac{\partial H_T}{\partial x} + v\frac{\partial H_T}{\partial y} \tag{8}$$

For the precision estimates in the first order, we apply standard linearized error propagation for each diagnostic variable, neglecting higher-order terms (covariances). For $w$, the sensitivities to the input parameters are derived from Eq. (7) as

$$\frac{\partial w}{\partial H_T} = -D; \ \frac{\partial w}{\partial D} = -H_T \tag{9}$$

Therefore, the precision in retrievals of $w \equiv w(H_T, D)$ from Equations (7) and (9) is





125
$$\sigma_w^2 = \left(\frac{\partial w}{\partial H_T}\sigma_{H_T}\right)^2 + \left(\frac{\partial w}{\partial D}\sigma_D\right)^2 = \left(D\sigma_{H_T}\right)^2 + (H\sigma_D)^2 \tag{10}$$

where $\sigma_D{}^2 = \sigma_{\partial u/\partial x}^2 + \sigma_{\partial v/\partial y}^2$

Meanwhile, to calculate the precision in the advection term $A$, we note from Eq. (9) that

$$\frac{\partial A}{\partial u} = \frac{\partial H_T}{\partial x}; \ \frac{\partial A}{\partial \left(\frac{\partial H_T}{\partial x}\right)} = u; \ \frac{\partial A}{\partial v} = \frac{\partial H_T}{\partial y}; \ \frac{\partial A}{\partial \left(\frac{\partial H_T}{\partial y}\right)} = v \tag{11}$$

and hence,

130
$$\sigma_A^2 = \left[(\partial_x H_T)^2 \sigma_u^2 + u^2 \sigma_{\partial_x H_T}^2 + \left(\partial_y H_T\right)^2 \sigma_v^2 + v^2 \sigma_{\partial_y H_T}^2\right] \tag{12}$$

Since from Eq. (9), $w_e = A - w$, the precision in $w_e$ retrievals is simply

$$\sigma_{w_e}^2 = \sigma_w^2 + \sigma_A^2 \tag{13}$$

To calculate the derivative uncertainties for any 2D field $f \in \{u, v, H_T\}$, along a given direction $(x, y)$, we combine two independent contributions:

135     a)    Instrument retrieval error for the given field, $f$ ($\sigma_f$), scaled by the effective grid-spacing, $\Delta x$ (20 km in our case), i.e.,

$$\sigma_{\partial f/\partial x}^{(1)} = \frac{\sigma_f}{\Delta x} \tag{14}$$

    b)    local derivative variability for the given field, $f$, computed as the standard deviation of $\partial f/\partial x$ within a window (0.6° in our calculations) centred around the grid-cell in consideration, i.e.,

$$\sigma_{\partial f/\partial x}^{(2)} = std\left[\left(\frac{\partial f}{\partial x}\right)_{\pm window}\right] \tag{15}$$

140   The total derivative uncertainty is the quadrature sum of these individual contributions,

$$\sigma_{\partial f/\partial x} = \sqrt{\left(\sigma_{\partial f/\partial x}^{(1)}\right)^2 + \left(\sigma_{\partial f/\partial x}^{(2)}\right)^2} \tag{16}$$

To estimate the accuracy (systematic uncertainty) of the retrievals in the first order, we consider the mean expected systematic offsets in the input parameters ($\Delta u, \Delta v, \Delta H_T, \Delta \partial_x H_T, \ \Delta \partial_y H_T, \Delta D$) from the known bias characteristics of MISR retrievals.





Thus, systematic uncertainty in $w$ from Eq. (7) is

$$\Delta w = \frac{\partial w}{\partial D} H_T + \frac{\partial w}{\partial H_T} D = -D.H_T - H_T.D \qquad (17)$$

with, $\Delta D = \Delta(\partial_x u) + \Delta(\partial_y v)$.

Similarly, the systematic uncertainty in $w_e$ from Eq. (8) is

$$\Delta w_e = \Delta A - \Delta w \qquad (18)$$

where

$$\Delta A = \Delta u \, (\partial_x H_T) + u \, \Delta(\partial_x H_T) + \Delta v \, (\partial_x H_T) + v \, \Delta(\partial_x H_T) \qquad (19)$$

*Scaling of Random Uncertainty with Sampling:* The retrieval technique is showcased in this study using a single instantaneous overpass, and hence part of the stated uncertainty reflects finite-sample ('sampling') uncertainty. Marine stratocumulus is organized on mesoscale cells (~20–50 km) with lifetimes of hours to days, hence scene-to-scene variability in cloud-top divergence and height advection is expected to be similar under similar forcing (Atkinson and Zhang, 1996; Wood and Hartmann, 2006). Terra/MISR samples scenes over a given geo-location at a fixed local time with a ~16-day repeat ground track and a ~380 km swath (Diner et al., 1998), so successive overpasses of a region are typically separated by far more than the integral time scale of the cloud field. It should be noted that the per-pixel uncertainties reported above are retrieval uncertainties only and do not include sampling uncertainty. Sampling uncertainty needs to be considered when spatial or temporal averages of the retrieved $w$ or $w_e$ fields are taken (e.g., a scene-mean or a regime composite). For any average $\bar{X}$ calculated over $N$ grid cells and/or scenes (which may or may not be all independent),

$$Var\,(\bar{X}) \approx \frac{\sigma_X^2}{N_{eff}} + Var_{sys} \qquad (20)$$

Here, $\sigma_X$ is the appropriate random uncertainty ($\sigma_w$ or $\sigma_{w_e}$; see Equations (10) and (13)), $N_{eff}$ is the effective number of independent samples, and $Var_{sys}$ represents systematic components of the precision budget that do not "average down" (e.g., height-assignment offsets, along-track anisotropy in MISR CMV). Thus, the sampling contribution to the standard uncertainty of a sample mean is

$$SE_{sampling}(\bar{X}) \approx \frac{\sigma_X}{\sqrt{N_{eff}}} \qquad (21)$$



For a single MISR overpass, $N_{eff}$ is set primarily by spatial autocorrelation over the 2D fields of cloud retrievals because Terra's revisit over a fixed region is ~16 days and each day's cloud field is typically a different realization. Hence, temporal autocorrelations (which would ideally have been another consideration) can be ignored in the case of long-term MISR means.

MISR's swath is long and relatively narrow (~380 km cross-track), so the number of spatial degrees of freedom depends on the analysed domain area $A$ and the field's spatial correlation area $A_c$. Following standard practice (e.g. Vallejos et al., 2014), the spatial degrees of freedom and by extension, effective sample size approximately reduces with spatial autocorrelation as

$$N_{eff;space} \approx \frac{A}{A_c}; A_c \approx \pi L_x L_y \tag{22}$$

where $L_x, L_y$ are integral spatial scales of the field (obtained from the area under the normalized autocorrelation function in
the zonal and meridional directions). Marine stratocumulus exhibits mesoscale cellular organization with typical correlation lengths of order 20–50 km (Atkinson and Zhang, 1996; Wood and Hartmann, 2006), but $L_x, L_y$ could also potentially be estimated from co-incident retrievals from MISR and MODIS multispectral cloud products.

For scene-mean estimates from a single overpass (such as in this study), the sampling term (from Eq. 24) is simply

$$SE_{sampling}(\overline{w}) \approx \frac{\sigma_w}{\sqrt{\dfrac{A}{\pi L_x L_y}}}; \ SE_{sampling}(\overline{w_e}) \approx \frac{\sigma_{w_e}}{\sqrt{\dfrac{A}{\pi L_x L_y}}} \tag{23}$$

**2.2 Datasets**

The proposed retrieval technique only utilizes data from MISR. Additional datasets are used for validation of MISR wind vectors and CTHs. Table 1 summarizes all data sources.

**2.2.1 MISR Cloud Motion Vectors (CMVs)**

The Multi-angle Imaging SpectroRadiometer (MISR) is a narrow-swath (~380 km) multi-angular imaging instrument with nine cameras pointed at nadir and view angles of ±70.5º, ±60.0º, ±45.6º, ±26.1º, and 0 (nadir) (Diner et al., 1998). We use the MISR Level 3 CMV monthly product (Version F02_0002, MI3MCMVN_002), which provides stereoscopically derived cloud-top heights (CTHs) and collocated horizontal winds $(u, v)$ at ~17.6 km resolution. Although MISR also provides higher-resolution CTH retrievals at 1.1 km, here we use CTHs at the same resolution as the motion vectors for consistency.

Cloud-motion vectors are derived by tracking features across MISR's multi-angle imagery. Triplets of cameras with asymmetric view geometry are used to disentangle apparent motion due to parallax from actual horizontal motion, enabling



simultaneous retrieval of wind and height (Moroney et al., 2002; Horváth, 2013). This retrieval is purely geometric and does not depend on ancillary thermodynamic profiles. As a result, MISR retrievals of cloud-top heights and winds have been demonstrated to be highly accurate and precise. A near-global comparison of MISR cloud-top heights against a space lidar

revealed a bias $\pm$ precision of -240$\pm$300 m for low clouds (Mitra et al., 2021). Meanwhile, depending on the method of validation, the zonal wind speed ($u$) was found to be nearly unbiased whereas the meridional wind speed ($v$) was shown to have a bias between -0.3 to -1.5 m s$^{-1}$, with precision typically in the range 2.4-3.7 m s$^{-1}$ (Horváth, 2013; Mueller et al., 2017). For this study, we shall use a nominal bias $\pm$ precision of $0 \pm 2.4$ m s$^{-1}$ for u and $-1.2 \pm 3.2$ m s$^{-1}$ for v. Leveraging the well-characterized nature of the error characteristics, Mitra et al. (2021) closed the error budget for MISR cloud-top heights and

motion-vector retrievals. As a result, we can propagate the known errors in MISR CMVs and CTHs to estimate retrieval errors in our retrievals of $w$ and $w_e$.

Unlike many monthly satellite products, MISR CMVs are not pre-gridded but stored as lists of individual retrievals, tagged by time, geolocation, and orbit. This allows use at the single-overpass level. Here, we subset one descending-node daytime overpass intersecting the stratocumulus deck off California (4 June 2018). Standard quality screening is applied (QA>50 or

high-confidence retrievals only), and we retain only low clouds (CTH<3 km) so that CMVs represent PBL-top winds.

### 2.2.2 ERA5 Reanalysis

The European Center for Medium-range Weather Forecasting (ECMWF) reanalysis model (ERA5; Hersbach et al., 2020) provides reference meteorological fields. Specifically, the profiles of horizontal winds ($u, v$), vertical pressure tendency ($\omega$, Pa s$^{-1}$), temperature, geopotential heights and water vapor mixing ratio along with surface air pressure, sea surface temperature

and boundary layer height (BLH) are utilized. The pressure levels were converted to geometric height using the geopotential heights for direct comparison with MISR heights.

ERA5 vertical pressure tendency is converted to mesoscale vertical air motion ($w$) following:

$$w = -\omega \frac{R_d T_v}{pg} \tag{24}$$

where, $T_v$ is the virtual temperature, $R_d$ is universal gas constant for dry air and $g$ is acceleration due to gravity. The ERA5 boundary-layer height (BLH) was used as a proxy for cloud top height. The ERA5 entrainment velocities were derived by

employing the mass-budget equation (5).

### 2.2.3 GOES (SatCORPS)

To provide an independent geostationary constraint, we use hourly cloud-top height (CTH) and cloud-top temperature (CTT) from the NASA Satellite ClOud and Radiation Property retrieval System (SatCORPS) dataset, which adapts CERES Edition

4 retrievals for GOES platforms (Minnis et al., 2020). For this study, GOES-15 (GOES-West) retrievals closest in time to the MISR overpass are collocated to the MISR grid and compared against MISR CTHs.



### 2.2.4 MODIS Terra Level-2 (MOD06)

To add a second estimate of CTH from a polar-orbiter, we use CTH estimates from the Moderate Resolution Infrared Spectroradiometer (MODIS; MOD06_L2), also onboard Terra. MODIS CTH for low clouds (CTH < 3 km) are always derived

based on conversion of retrieved infrared cloud-top temperatures (Platnick et al., 2017). MODIS CTHs are also collocated to the MISR grid to compare against the MISR estimates of CTH.

**Table 1:** Details of Data Sources Used in Study

| Dataset | File Name & Version | Resolution | Variables Used | Uncertainty |
|---|---|---|---|---|
| **MISR CMV** | MISR Level 3 Cloud Motion Vector Monthly Product V002 (MI3MCMVN_002) | Monthly (with tables of successful retrievals; not gridded); 17.6 km | Cloud-Motion Vector (CMV) wind velocities | u: $0\pm2.4$ m s$^{-1}$ v: $-1.2\pm3.2$ m s$^{-1}$ (Horváth, 2013; Mueller et al., 2017) |
| | | | Stereo Cloud-Top Heights (CTH) | $-240\pm300$ m (Mitra et al., 2021) |
| **ERA5** | 5$^{th}$ Generation European Center for Medium-range Weather Forecasting (ECMWF) reanalysis model output | Hourly; 0.25° | Multi-Level u, v, Temperature, Water-Vapor Mixing Ratio, Pressure Tendency, Geopotential Heights | N/A |
| | | | Single-level SST, PBL heights, Surface Pressure | |
| **SATCORPS GOES** | SatCORPS CERES GEO Edition 4 GOES-16 Northern Hemisphere Version 1.4 (CER_GEO_Ed4_GOE15_NH_V01.2) | Hourly; 8-km | Low-cloud cloud-top heights & temperatures | Nominal uncertainty of GOES CTH ~500 m |
| **Terra MODIS** | MODIS/Terra Calibrated Radiances Level 1B (MOD021KM) | 5-minute Granule; 1-km | Calibrated RGB radiances | N/A |
| | MODIS/Terra Cloud Product Level 2 (MOD06_L2) | | Low-cloud cloud-top heights | $60\pm660$ m (Mitra et al., 2021) |
| | MODIS/Terra Geolocation Level 2 (MOD03) | | MODIS retrieval geolocations | N/A |



*Algorithmic Implementation:* All datasets are mapped to a regular latitude–longitude grid spanning the MISR swath, with 0.2° spacing in both latitude and longitude. This corresponds to ~22 km meridionally and ~19–20 km zonally over the subtropical oceans that are the area of interest and is comparable to the native ~17.6 km MISR sampling. $H_T$, $u$, and $v$ from the native MISR grid are linearly interpolated onto this mesh with grid cells lying farther than 0.2° from the nearest MISR retrieval left undefined to avoid extrapolation into data gaps.

Spatial derivatives are computed in metric space using great-circle distances. For each target grid cell (blue circle in Fig. 1), we gather all valid retrieval pairs that straddle the cell in the north–south direction and, separately, in the east–west direction within the analysis window appropriate to the diagnostic. For each pair, we calculate gradients as the difference in the field values divided by the great-circle separation along the meridional (N–S) or the zonal (E–W) directions, and the directional gradient is calculated as the arithmetic mean of these pairwise slopes. This extreme-pair approach is robust to swath gaps and

returns $\frac{\partial (.)}{\partial x}$ and $\frac{\partial (.)}{\partial y}$ in native units per kilometre and the derivatives are undefined when the required neighbour pairs are absent.

For divergence, a half-width of 0.4° stabilizes $\frac{\partial u}{\partial x}$ and $\frac{\partial v}{\partial y}$ and yields a reliable $\nabla_h . \overrightarrow{V_h}$ for equation (1). For the height-advection term $u \frac{\partial H_T}{\partial x} + v \frac{\partial H_T}{\partial y}$, a half-width of 0.2° preserves mesoscale structure in $H_T$. Units are kept consistent (either convert $H_T$ to metres or the gradients to metres$^{-1}$), so that the final $w$ and $w_e$ retrievals are in m s$^{-1}$, which are finally reported in units of cm s$^{-1}$.

Because a single MISR overpass (which provides the snapshot from which our retrievals are made) can result in small-scale variability in $w$, the entrainment calculation replaces the pointwise $w$ in equation (5) by a local environmental mean,

$$\langle w \rangle_\rho (x, y) = \frac{1}{N_\rho (x,y)} \sum_{d \leq \rho} w (x', y') \tag{25}$$

where the average is taken over all valid neighbours within a radius $\rho = 0.4°$ (great-circle metric). The operational entrainment used for this case is therefore derived from equation 5 as

$$w_e (x, y) = u \frac{\partial H_T}{\partial x} + v \frac{\partial H_T}{\partial y} - \langle w \rangle_\rho (x, y) \tag{26}$$




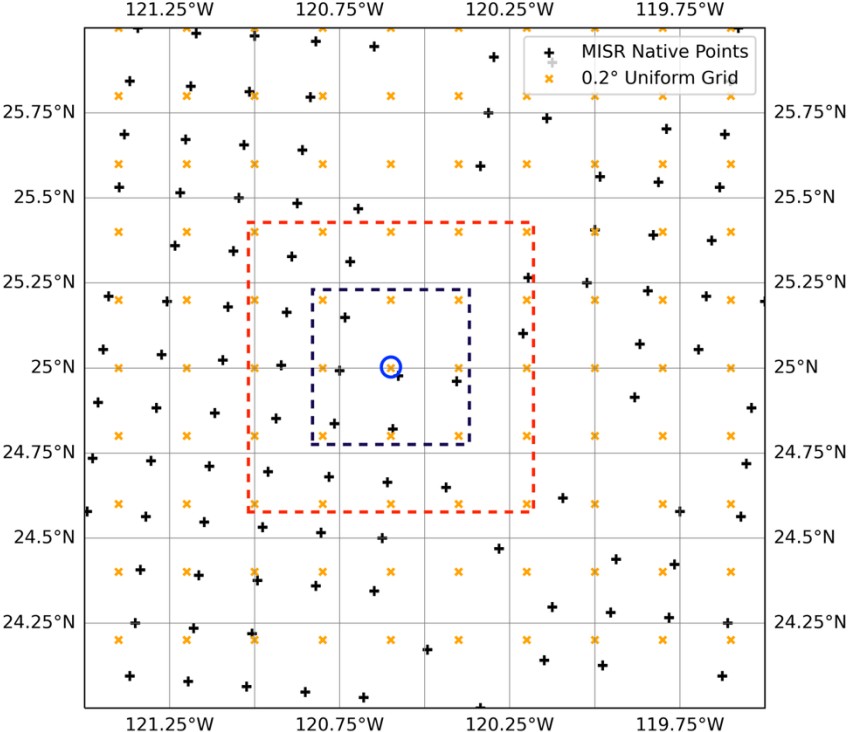

**Figure 1**: A zoomed-in view of the spatial distribution of the native MISR data grid (black plus signs) and the grid centres of the uniform 0.2° grid to which all MISR data is resampled (orange crosses). Also shown are the windows used to calculate derivatives for the estimation of vertical air velocity (dashed red) and entrainment rate (dashed black) at cloud-top for a target grid-cell (blue circle).

All outputs ($\frac{\partial u}{\partial x}, \frac{\partial v}{\partial y}, \frac{\partial H}{\partial x}, \frac{\partial H}{\partial y}, w, w_e$ and the advective term $u \frac{\partial H_T}{\partial x} + v \frac{\partial H_T}{\partial y}$) are produced on the regular 0.2° mesh to which all input data had been resampled; grid cells lacking sufficient neighbors for the directional pairs or the local average remain undefined.

## 3. Results

We compare MISR wind vectors with collocated ERA5 fields and MISR CTH with independent CTH estimates from GOES–SatCORPS and MODIS, as well as ERA5 BLH. We then present MISR-derived $w$ and $w_e$, evaluate them against equivalent diagnostics from ERA5, and quantify retrieval uncertainties. Intercomparisons of input variables are conducted at the native MISR CMV resolution (~17.6 km), while derived fields are produced on the uniform 0.2°-resolution grid.



### 3.1 Case Description and Intercomparison

We analyse a single Terra overpass on 4 June 2018 sampling an unbroken marine stratocumulus deck off the California coast. All datasets (MISR CMV, ERA5, GOES–SatCORPS, MODIS) are restricted to this day and collocated to the MISR footprint. MISR retrievals are filtered to QA > 50 (high confidence retrievals only) and CTH < 3 km.

Figure 2a shows continuous low cloud with embedded cellular texture in GOES visible reflectance. MISR stereo CTH estimates were available for the whole sector, with few gaps due to cloud clearing and presence of cirrus clouds. As MISR
reports the CTH above the mean sea level, some of the cloud top heights could be negative. In fact, the acceptable values of MISR CTH in the operational MISR cloud-top height retrievals are -500 m to 20 km above mean sea level (with negative heights signifying cloud-top height retrievals below mean sea-level). Only 8 negative CTH values were present within this scene (all between -100 m to 0) and hence were neglected in further analysis. However negative CTH will need to be converted to heights over the geoid for retrieval calculations in further iterations of this technique. The CTH increased from North to the
South, with East-West changes being both positive and negative. The co-located MISR CMVs (Fig. 2b) depict a coherent North-westerly flow with gentle but organized gradients at the CMV resolution. The winds were north-westerly on the western end of the swath, and westerly on the eastern end North of 27.5° N. To the South of 27.5° N, the winds were north-westerly on the western end of the swath, and almost northerly at the eastern edge. This wind patterns and cloud top height changes are consistent with the higher-pressure system and high low-level cloudiness as over the Northeast Pacific as reported by Wood
and Bretherton (2006). ERA5 boundary-layer heights (BLH) and winds for the overpass time (Fig. 2c) present a broadly consistent flow and a slowly varying inversion depth along the swath. However, around 30° N and -118.5° W, the ERA5 winds were north-westerly, as opposed to the MISR winds that were almost westerly.



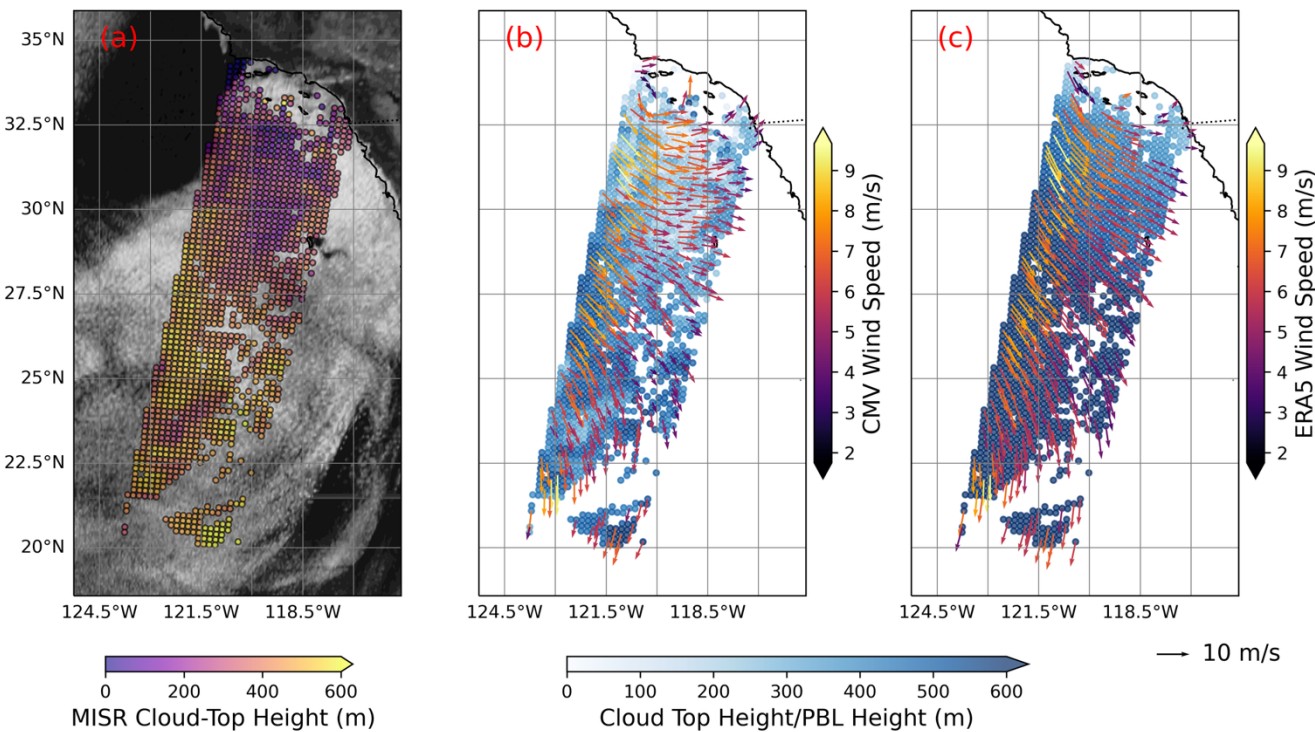

**Figure 2**: (a) GOES visible channel reflectance (greyscale) and MISR CTH for low clouds (colours), (b) MISR CTH (blue-scale colours) and MISR cloud-motion vectors (red scaled coloured arrows) and (c) ERA5 PBL depths (blue-scale colours) and ERA5 horizontal wind vectors (red scaled coloured arrows).

The MODIS visible satellite imagery shows similar patterns as GOES, showcasing widespread stratus coverage over the region (Figure 3a). Due to higher spatial resolution, the cellularity of marine stratocumulus clouds is nicely visible in the MODIS imagery. Thin upper-level clouds are present near the coast as reported by the MODIS cloud top heights (Fig. 3b). Also visible are rectangular patches in the MODIS low cloud top height estimates. These are related to empirical adjustments made over discrete 1° latitude-longitude regions while converting MODIS retrieved low cloud-top temperatures to corresponding cloud-top heights (elaborated further while discussing Fig. 4).

In multilayer cloud scenes (e.g. cirrus over stratus), MISR's geometric stereo retrieves the lower cloud properties when the overlying cirrus is optically thin. Comparisons against space-borne lidar has shown a robust low-cloud detection by MISR for cirrus optical depth $\tau \lesssim$ 0.3-0.4 (Mitra et al., 2021). Moreover, whenever the low cloud is detected, the accuracy of the retrieved low-cloud CTH is not degraded by the thin cirrus above (Mitra et al., 2021; Mitra et al., 2023). Conversely, for thicker cirrus $\tau > 0.5$, MISR stereo retrieves cirrus top heights instead, and hence can be neglected from further low cloud analysis. This clear distinction in low versus high cloud retrievals of CTH within the MISR record is a distinct advantage over infrared techniques employed in other satellite sensors (GOES, MODIS), which often results in spurious "mid-level" cloud-height



retrievals for multi-layered scenes. The data points that are not plotted along the MISR track plots in Fig. 1a are due to a mix of such cirrus contamination or low MISR CMV QA values.

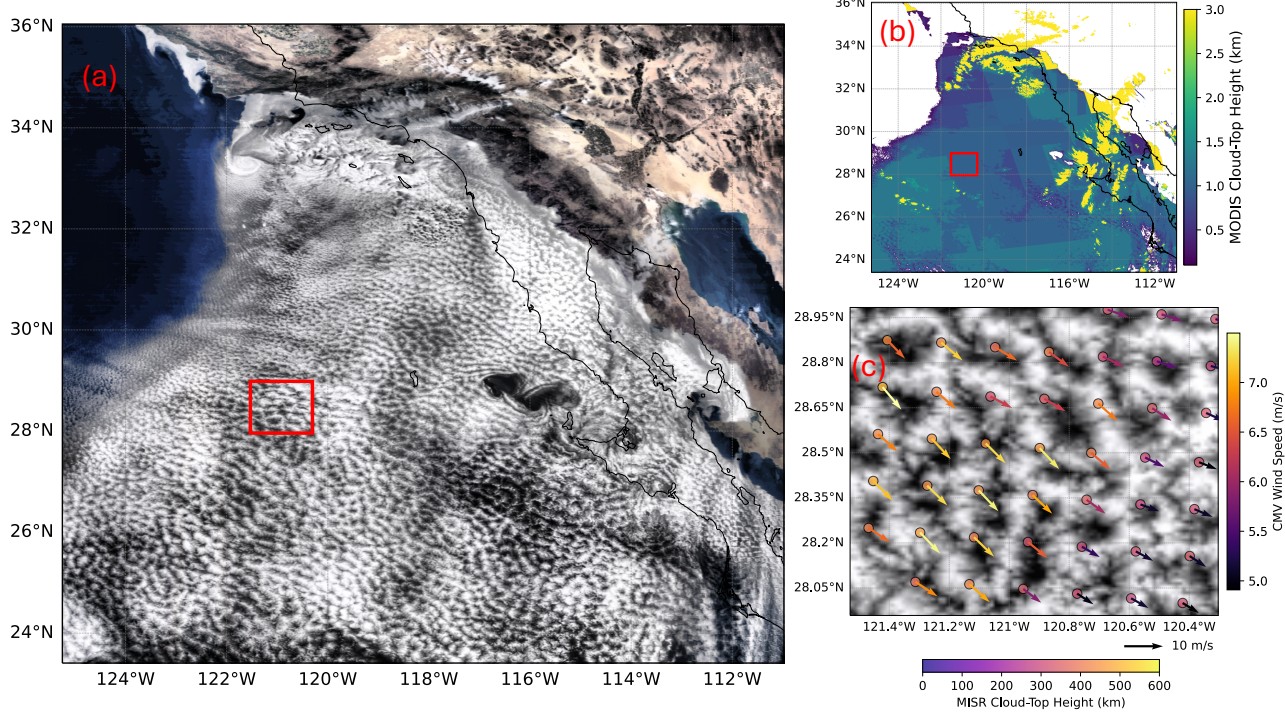

**Figure 3**: (a) MODIS True colour RGB imagery, (b) MODIS Cloud-top heights and (c) MODIS visible imagery over a small
subset over the case study area with inlaid MISR cloud-top heights (coloured circles) and cloud-motion vectors (coloured arrows). In panels (a) and (b), the subset region shown in panel (c) is highlighted with a red box. To aid interpretation, panels (a) and (b) have been contrast-enhanced using luminance-based stretching and local equalisation to highlight the mesoscale cellular structures in the cloud field.

Zooming into a subset of the field of MISR CTH and CMV wind vector retrievals against the backdrop of MODIS true colour imagery (Fig. 3c), individual cellular elements of the cloud structure can be visually resolved. MISR vectors delineate confluence around cell edges, i.e., the spatial structure that makes entrainment diagnosis meaningful at ~17.6 km. Over this 1°x1° region, wind speed changes of ~2 m s$^{-1}$, and cloud top height changes of ~300 m occurred. This further reinforces the ability of MISR CMV to capture the mesoscale variability in cloud and dynamical properties of marine stratocumulus decks.

Accurate cloud-top height (CTH) is a primary requirement for this retrieval framework. Figure 4 shows comparison of MISR CTH with independent estimates from GOES–SatCORPS and MODIS, and with the ERA5 boundary-layer height (BLH). We also include an empirical CTH derived from GOES cloud top temperature (CTT) and ERA5 sea surface temperature (SST) using the correction proposed by Painemal et al. (2013):



$$CTH = \frac{SST - CTT + 1.35}{0.0095} \qquad (27)$$

This relationship was developed over southeast Pacific stratocumulus to mitigate errors introduced by lapse-rate assumptions when converting CTT to height.

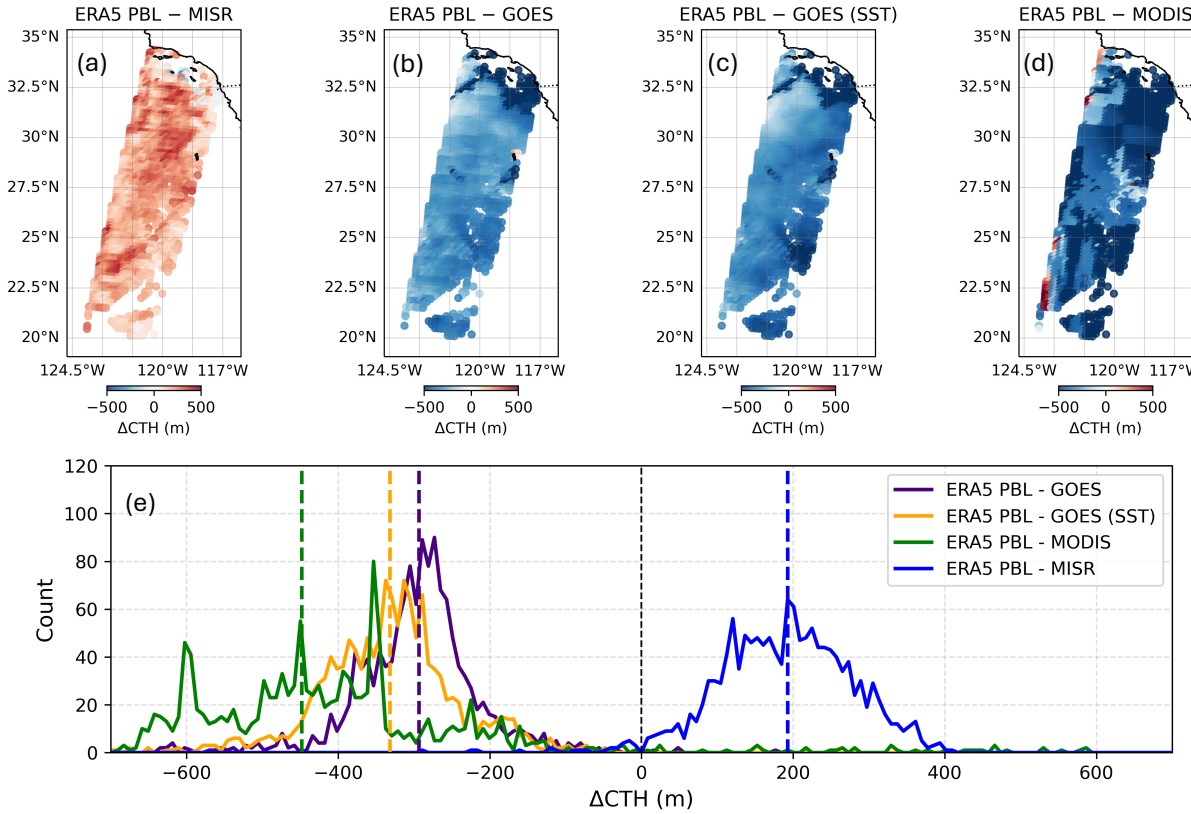

**Figure 4**: Differences between (a) ERA5 PBL depths and MISR CTH, (b) ERA5 PBL depths and GOES CTH, (c) ERA5 PBL depths and GOES-SST CTH and (d) ERA5 PBL depths and MODIS CTH. (e) Histograms of the differences between ERA5 PBL depths with MISR CTH (blue), GOES CTH (purple), GOES (SST) CTH (orange) and MODIS CTH (green). Mean differences for each histogram are depicted as dashed vertical lines of the corresponding colour.

Mean differences (ERA5 PBLH − CTH) over the domain are $+188.9 \pm 87.3$ m (MISR), $-343.2 \pm 282.2$ m (GOES), $-359.2 \pm 199.2$ m (GOES-SST) and $-608.2 \pm 1037.0$ m (MODIS). The signs and spreads are consistent with the expected behaviour of the retrievals. Thermal-infrared methods (GOES, MODIS) infer heights from brightness temperatures using ancillary profiles that may not capture sharp, shallow inversions; when the inversion is strong, the radiances constrain cloud-top temperature poorly and heights tend to be biased high. Their errors also depend on low-cloud optical depth and temperature contrast, yielding broad distributions even within a single MISR scene (Mitra et al., 2021). MODIS operational adjustments based on climatological lapse rates reduce long-term biases but do not guarantee scene-specific accuracy, which is consistent with the






larger spread and the 1° x 1° block-like discontinuities evident in Fig. 3b. Similar to MODIS, GOES CTHs exceed ERA5 BLH and MISR CTH; applying the GOES–SST regression further increases the overestimate in this case, suggesting that factors beyond SST (e.g., departures from a well-mixed boundary layer or regional tuning of the coefficients) contribute to the
mismatch. By contrast, MISR uses geometric stereo to triangulate feature heights. In marine stratocumulus the tracked features often sit slightly below the radiative top, so MISR CTH is typically below the inversion, yielding a small positive value of ERA5 PBLH – MISR CTH. Critically, this offset varies far less with optical depth than in IR methods (Mitra et al., 2021; Loveridge and Di Girolamo, 2025), producing a tighter distribution that is preferable when propagating height errors into the uncertainties in the retrieval of the vertical air motion and entrainment velocities.

We compare MISR CMV winds with ERA5 at two levels: (i) the ERA5 model level closest to the MISR CTH and (ii) the ERA5 planetary boundary-layer height (PBLH). Figure 5 summarizes the differences. Sampling ERA5 at MISR CTH yields domain-mean ($\pm 1\sigma$) differences of $\Delta u$ = +0.01$\pm$0.75 m s$^{-1}$, $\Delta v$ = +2.17$\pm$1.19 m s$^{-1}$ and $\Delta |V|$ = -1.42 $\pm$1.11 m s$^{-1}$ (MISR–ERA5). Sampling at ERA5 PBLH reduces the MISR–ERA5 speed bias to -1.17 $\pm$1.20 m s$^{-1}$, while leaving the components similar ($\Delta u$ = +0.09$\pm$0.77 m s$^{-1}$, $\Delta v$ = +2.12$\pm$1.24 m s$^{-1}$). The ~0.25 m s$^{-1}$ improvement in speed bias is expected because
height-assignment error projects onto winds through vertical shear. The first-order decomposition of wind-speed errors between two independent datasets can be written as:

$$\Delta V = \overbrace{\left(\frac{\partial V}{\partial z}\right)\Delta z}^{\text{height assignment}} + \overbrace{\left(\frac{\partial V}{\partial t}\right)\Delta t}^{\text{temporal mismatch}} + \overbrace{V_{\text{rep}}}^{\text{spatial/scale mismatch}} + \overbrace{\varepsilon_{\text{alg}}}^{\text{retrieval \& collocation}} \qquad (28)$$

where $\Delta z$ is the mean offset between the true inversion height and level at which winds are retrieved. Moving from MISR CTH to ERA5 PBLH reduces $\Delta z$ (Fig. 3), so the shear term $(\partial V/\partial z)\Delta z$ shrinks and the speed bias decreases accordingly. That
height-to-wind mapping is well-established as height assignment is often the dominant AMV error source and projects into vector differences through vertical shear (e.g., Velden and Bedka, 2009; Salonen et al., 2015; Córdoba et al., 2017).

The component asymmetry in this case, near-zero mean $\Delta u$ but a persistent positive $\Delta v$ of about 2 m s$^{-1}$ reflects MISR's viewing geometry and error anisotropy. On Terra's descending node, the along-track direction is approximately meridional, so the MISR $v$ wind component maps closely to along-track motion, while $u$ is predominantly cross-track. MISR's CMV
algorithm derives motion from fore–nadir–aft image image triplets separated by ~3.5 min. The component errors are known to be larger in the along-track than the cross-track direction and to covary with stereo height errors because the same parallax geometry underpins both (Horváth, 2013). Residual geo-registration and co-registration artefacts can further introduce small, direction-dependent biases (e.g., noted in MISR–Meteosat intercomparisons; Lonitz and Horváth, 2011). These retrieval-intrinsic effects add to temporal offsets and representativeness differences, for example MISR provides instantaneous ~17.6
km wind vectors at cloud top, whereas ERA5 wind estimates are an hourly ~31 km analysis (e.g., Janjić et al., 2018). Together,



these terms explain why the speed bias improves when sampling ERA5 at PBLH, yet the meridional component difference remains the dominant residual.

**Figure 5**: (a) Differences between MISR and ERA5 wind speeds at MISR CTH. Histograms of the differences between MISR and ERA5 (b) u-component of wind velocity, (c) v-component of wind velocity and (d) wind speeds at MISR CTH (blue) and





at ERA5 PBL top (dashed black). For the histograms in panels (b), (c) and (d), all units are m s⁻¹ and the mean differences are denoted by vertical lines with the same style and colour of the corresponding histograms.

Crucially, this interpretation does not require invoking a global calibration error in MISR vectors. Rather, the data are consistent with (i) a small height-assignment contribution that diminishes when the comparison level better matches the
inversion, plus (ii) anisotropic along-track uncertainties intrinsic to the stereo-tracking geometry, and iii) temporal and spatial scale mismatches between an instantaneous stereo observation and an hourly coarser reanalysis model.

In summary, the 4 June 2018 scene is a low-topped, cellular stratocumulus deck in which MISR stereo cloud top heights and winds resolve mesoscale strain and confluence at ~20 km scales, enabling stable wind divergence and entrainment retrievals.

### 3.2 Retrievals

We retrieve mesoscale cloud-top vertical air velocity ($w$) and mesoscale cloud-top entrainment rates ($w_e$) using both MISR CMV $u$ and $v$ at MISR CTH and for ERA5 $u$ and $v$ at ERA5 BLH, for the 4 June 2018 scene. All input parameters were re-gridded to and the retrievals made at a uniform 0.2° grid. During the retrievals, the height-advection term ($u\frac{\partial H}{\partial x} + v\frac{\partial H}{\partial y}$) is also output for comparative analysis against the retrieved $w$ and $w_e$. Figure 6 summarizes the cloud-top vertical air velocity $w$, the entrainment velocity $w_e$, and the height-advection term for MISR, ERA5, and their differences (MISR-ERA5). The portion of
the swath shown originally comprised of 1397 MISR data points, out of which $w$ and $w_e$ retrieval was made for 1312 (~ 94 %) and 1336 (~ 96%) points, respectively. The differences in the number of valid retrievals is due to the different window sizes used to calculate their underlying derivatives (Figure 1).

MISR $w$ (Fig. 6a) exhibits a clear meridional dipole near the coast with intricate mesoscale structure at the same resolution as the retrieval (~0.2°). There is markedly stronger descent near the coast, followed by a region of ascent centred around 30° N,
followed by a region of weaker descent dotted with patches of very weak ascent. The domain mean and standard deviation are -0.36±0.65 cm s⁻¹. Meanwhile, ERA5 $w$ (Fig. 6b) shows a broadly similar large-scale pattern but is smoother in its features and extents of ascent-descent regions. The domain mean and standard deviation are -0.22±0.45 cm s⁻¹. The difference of MISR and ERA5 $w$ (Fig. 6c) is largely of the same pattern as that of MISR $w$ (Fig. 6a). This is likely because MISR retrievals of CTH carry more spatial structure than does the corresponding field of ERA5 PBL heights (Fig. 2b and Fig. 2c), which resolves
cell-scale convergence/divergence that the smoother underlying fields of ERA5 filters. The domain mean and standard deviation of differences in ERA5 and MISR derived mesoscale vertical air motion at the cloud top is 0.14±0.73 cm s⁻¹.




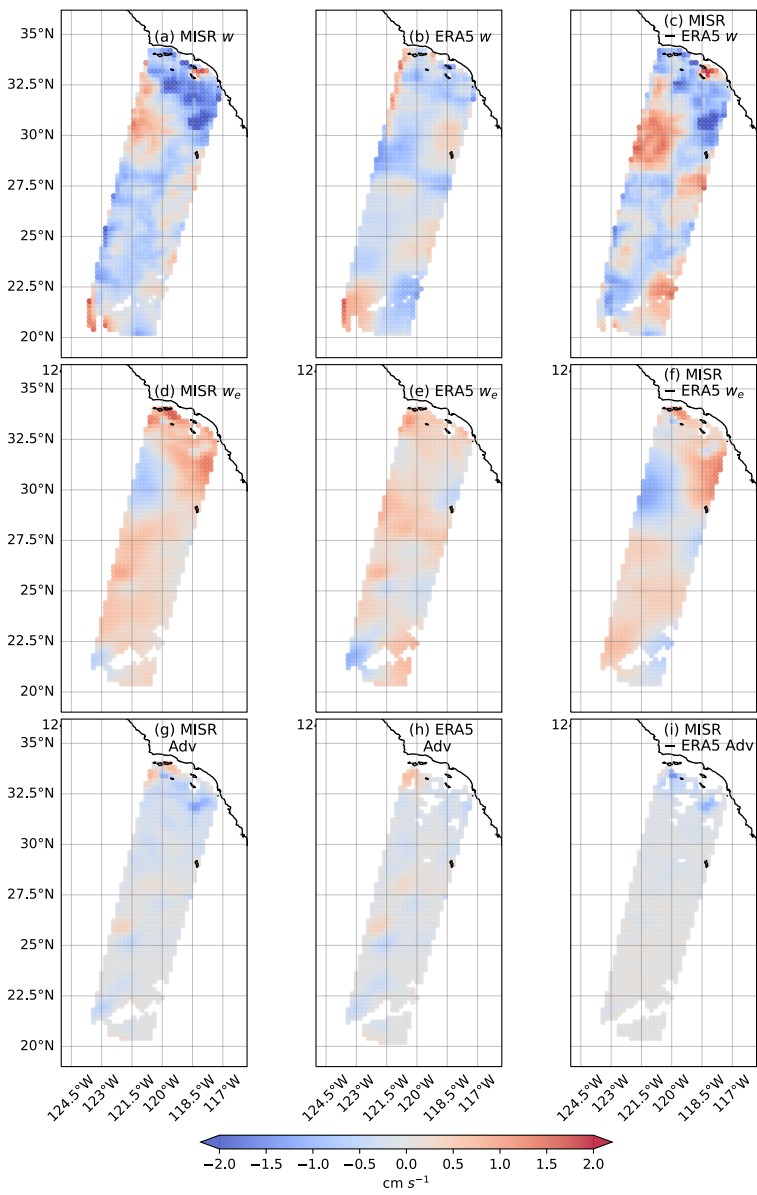

**Figure 6**: (a) MISR $w$ retrievals (b) ERA5 $w$ (c) Differences between MISR and ERA5 $w$ (d) MISR $w_e$ retrievals (e) ERA5 $w_e$ (f) Differences between MISR and ERA5 $w_e$ (g) MISR CTH advective term (e) ERA5 PBL height advective term (f) Differences between MISR and ERA5 PBL advective terms.

MISR $w_e$ (Fig. 6d) is predominantly positive across the deck with a gentle NE-SW gradient; the domain mean and standard deviation are $0.28\pm0.40$ cm s$^{-1}$. Meanwhile ERA5 $w_e$ (Fig. 6d) is also predominantly positive with a domain mean and standard deviation are $0.19\pm0.33$ cm s$^{-1}$. Since the ERA5 retrieved $w$ near the coast is not as negative as that from MISR, ERA5 retrieved $w_e$ near the coast is not as positive as that from MISR. Moreover, the locations of negative values of MISR



and ERA5 $w_e$ are dissimilar in places. However, their MISR-ERA5 $w_e$ differences (Fig. 6e) are typically low at such locations, and the differences are likely much lower than random uncertainties associated with these retrievals. The domain mean and standard deviation of ERA5 and MISR cloud-top entrainment rates is -0.09±0.46 cm s$^{-1}$ (also, Fig. 7b).

The advection term A ($u\frac{\partial H_T}{\partial x} + v\frac{\partial H_T}{\partial y}$) is weak in both products (Figs. 6g-h) and its difference is near zero over most of the scene (Fig. 6i), with a domain statistic of -0.09 ± 0.19 cm s$^{-1}$. Thus, for this case, differences in the retrieved advective

component contributes little to the MISR-ERA5 mismatch in $w_e$, which is largely driven by the differences in retrieved values of $w$. Which, in turn, reflects the greater spatial heterogeneity in MISR CTH relative to ERA5 BLH. From Eq. 8, $w_e = A - w$ (positive $w$ upward). Therefore, for samples with upward mesoscale vertical motion ( $w > 0$ ), obtaining positive entrainment requires the height-advection term to exceed the upward motion. When $A$ is small, positive $w_e$ tends to occur where $w$ is weakly negative (subsiding).

The statistics of the overall distributions of the ERA5 and MISR derived $w$ and $w_e$ for the case are depicted in Figure 7. We quantify similarity between the ERA5 and MISR histograms by the area of overlap between their respective normalized probability densities (0-1 with 1 identical). The distribution of ERA5 and MISR $w$ (Fig. 7a) for the case was largely similar (79% overlap), with the MISR reported $w$ having a longer negative tail (stronger subsidence). The percentage of retrievals of descent (negative $w$) is slightly greater in MISR (982; 75%) than ERA5 (959; 73%). Overall, the percentage of retrievals with

descent stronger than 2 cm s$^{-1}$ (i.e., $w$ < -2 cm s$^{-1}$) is 13% in MISR and 3% in ERA5. Moreover, the distribution of ERA5 and MISR $w_e$ (Fig. 7b) for the case was largely similar (81% overlap), with the MISR derived $w_e$ having a longer positive tail (stronger cloud-top entrainment). The percentage of retrievals with valid cloud-top entrainment or positive $w_e$ is greater in MISR (950; 71%) than ERA5 (867; 65%). Overall, the percentage of retrievals with cloud-top entrainment stronger than 0.5 cm s$^{-1}$ (i.e., $w_e$ > 0.5 cm s$^{-1}$) is 28% in MISR and 17% in ERA5.

Although ERA5 and MISR retrievals of $w$ and $w_e$ show a linear relationship (Figures 7c and 7d), there are enough high sample-level differences that result in poor correlation between the two pairs of retrievals. The coefficient of linear correlation between MISR and ERA5 $w$ is 0.2 and between MISR and ERA5 $w_e$ is 0.25. Differences between ERA5 and MISR $w$ are found to lie within ±0.25 cm s$^{-1}$ for ~25% of all retrievals, while differences between ERA5 and MISR $w_e$ are found to lie within ±0.25 cm s$^{-1}$ for ~32% of all retrievals.



**Figure 7:** (a) Histograms of MISR (green), ERA5 (orange) and differences in MISR and ERA5 (purple) $w$ (b) Histograms of MISR (green), ERA5 (orange) and differences in MISR and ERA5 (purple) $w_e$ (c) Differences of MISR and ERA5 $w$ as a joint distribution of ERA5 and MISR $w$ (d) Differences of MISR and ERA5 $w_e$ as a joint distribution of ERA5 and MISR $w_e$ (e) Differences of MISR and ERA5 wind speeds at ERA5 PBL heights ($V$) as a joint distribution of ERA5 and MISR $w$ (f) Differences of MISR and ERA5 wind speeds at ERA5 PBL heights ($V$) as a joint distribution of ERA5 and MISR $w_e$ (g)





Differences of MISR CTH and ERA5 PBL heights as a joint distribution of ERA5 and MISR $w$ (h) Differences of MISR CTH and ERA5 PBL heights as a joint distribution of ERA5 and MISR $w_e$.

Figures 7e, 7f, 7g, 7h show that there is no strong systematic relationship between the point differences in the input parameters
(i.e., wind speeds and cloud-top heights) and the differences in the retrievals from ERA5 and MISR. This suggests that the point differences in retrieved estimates of $w$ and $w_e$ from MISR and ERA5 likely depend strongly on the spatial patterns in the 2D field of the input parameters and not the underlying point estimates of the input parameters.

An interesting subset of points (N=32) in these figures are the samples for which MISR-ERA5 cloud-top wind speed differences are negative and absolute CTH differences are less than 50 m (darker colors in the bottom left quadrant of Figures
7e and 7g and in the top right corners of Figures 7f and 7h). Unlike these points, MISR estimates of CTHs and winds are typically lower than respective ERA5 estimates (Figures 4 and 5). The mean difference in MISR and ERA5 CTHs for these points are 51±58 m and the mean difference in MISR and ERA5 wind speeds at CTH are -2.1±2.4 m s$^{-1}$. The majority (25 out of 32) of these points result in negative $w$ and positive $w_e$ retrievals from both MISR and ERA5. For these points, the mean difference between MISR and ERA5 retrievals of $w$ is -0.5±0.5 cm s$^{-1}$ and the mean difference between MISR and ERA5
retrievals of $w_e$ is -0.25±0.33 cm s$^{-1}$. Thus, close agreement in pointwise cloud-top height does not eliminate the systematic wind-speed difference; it persists (and can even increase) because part of the MISR wind-component uncertainty is directionally anisotropic (along-track vs cross-track) and does not vanish with improved height co-location (see Sect. 3.1).

With scene-mean input uncertainties $\sigma_u$ = 2.4 m s$^{-1}$, $\sigma_v$ = 3.2 m s$^{-1}$ and $\sigma_H$ = 300 m (and the windowed derivatives described in Section 2.1), random uncertainties for each retrieval point are propagated using Equations 10-13 and systematic uncertainties
are propagated using Equations 17-19. Mean propagated systemic uncertainty in $w$ and $w_e$ (i.e., $\Delta_w, \Delta_{w_e}$) are found to be -0.6 cm s$^{-1}$ and -0.4 cm$^{-1}$. The mean random uncertainty in in $w$ and $w_e$ (i.e., $\sigma_w, \sigma_{w_e}$) are found to be 0.7 and 0.5 cm s$^{-1}$, respectively. Fractional uncertainties ($\sigma_w/|w|$) are calculated from these estimates of random uncertainty and their relationship with the underlying CTH is studied in Figure 8. As expected, for very low values of cloud-top vertical velocity and entrainment velocities ($w, w_e \sim 0$), the fractional uncertainty can be asymptotically large.





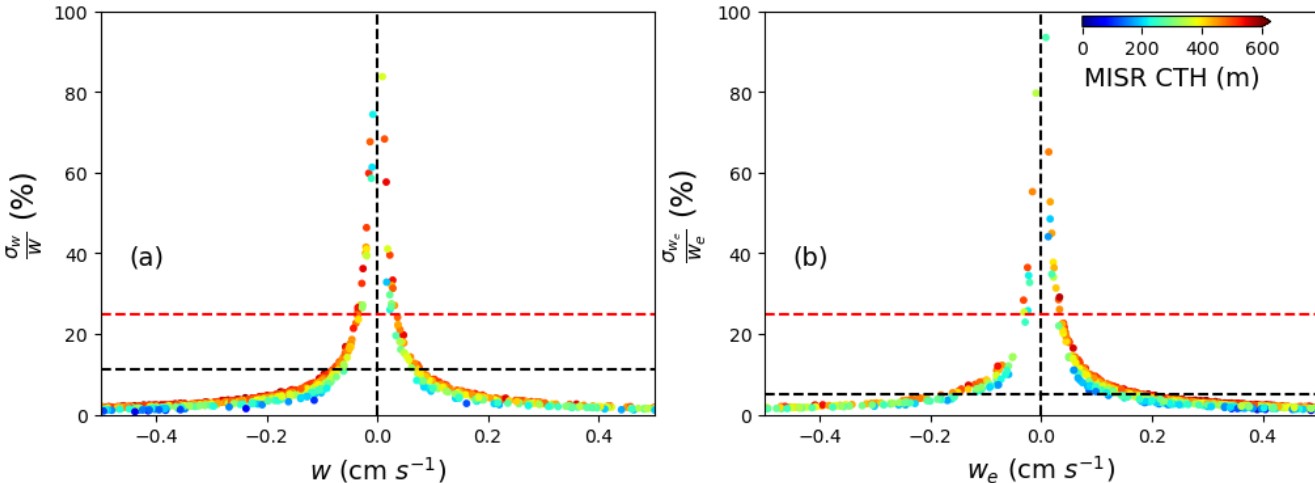


**Figure 8:** (a) Pointwise fractional uncertainty of MISR-based retrievals of $w$ (i.e., $\sigma_w/|w|$) against the retrieved $w$. (b) Pointwise fractional uncertainty of MISR-based retrievals of $w_e$ (i.e., $\sigma_{w_e}/|w_e|$) against the retrieved $w_e$. For both panels, black dashed lines show the domain-mean fractional uncertainties; red dashed lines mark a 25% benchmark. The sharp rise near $w, w_e \sim 0$ reflects division by small magnitudes.

Based on visual examination of the distributions of fractional uncertainty (Fig. 8), an ad hoc threshold of 25% fractional uncertainty is chosen for "physically meaningful" retrievals. For the present scene, fractional uncertainties lower than the 25% threshold are found for 1268 (i.e., 97%) points for $w$ retrievals and in 1122 (i.e., 85%) points for $w_e$ retrievals. Mean fractional uncertainty over all points are calculated to be 11% for $w$ retrievals and 5% for $w_e$ retrievals.

As noted in Section 2.2.1, the estimates of $\sigma_w, \sigma_{w_e}$ are merely retrieval uncertainties, which scale with successive independent 465 sampling. For the scene considered, using typical mesoscale lengths from the literature (e.g., $L_x = L_y = 40$ km; Wood and Hartmann, 2006), the correlation area is estimated as $A_c \approx \pi \times 40 \times 40 \ km \approx 5000 \ km^2$. The analyzed domain in our study covers an area approximately $2.0 \times 10^5 \ km^2$, hence, $N_{eff;space} \approx 40$, and the sampling standard errors for scene-means (from Eq. 25) are

$$SE_{sampling}(\overline{w}) \approx \frac{0.7}{\sqrt{40}} = 0.11 \ cm \ s^{-1}; \ SE_{sampling}(\overline{w_e}) \approx \frac{0.5}{\sqrt{40}} = 0.08 \ cm \ s^{-1}$$

These numbers are merely representative and are subject to change for actual values of $A, L_x, L_y$. As more scenes are aggregated within a homogeneous regime (e.g., same region/season and similar synoptic background), the random component of the mean retrieval uncertainty narrows roughly by a factor of $1\big/\sqrt{N_{eff}}$. A suitable averaging strategy to improve sampling precision for future atmospheric process-based analyses can focus on regime-based aggregation of multiple overpasses from the same season and synoptic class (e.g., similar lower-tropospheric stability, SST, and PBL depth).



However, it should also be noted that the scaling of random errors with $1\big/\sqrt{N_{eff}}$ does not happen indefinitely. This is because systematic contributors to the MISR precision budgets – such as uncertainties arising from discrete pixel registration or along-track anisotropy in retrieved wind components – do not average down and can only be treated by instrument-level calibration or explicit algorithmic improvements. Estimating this irreducible component robustly requires an independent reference (e.g., aircraft or surface-based observations) across multiple independent scenes within a given weather regime.

**4. Summary & Discussions**

Entrainment of warm dry air from above the cloud layer into the cloud is a critical process that modulates microphysical and radiative properties and, in turn, cloud lifetime (Wood, 2012; Bretherton et al., 2007; Mellado, 2017). Here we show that cloud-top vertical velocity $w$ and entrainment velocity $w_e$ can be retrieved from MISR alone using stereoscopic cloud-top height $H_T$ and cloud-motion vectors $(u, v)$ retrievals. As MISR provides co-located heights and winds at ~17 km resolution,

the retrievals correspond to meso- spatial scales, the scale at which stratocumulus organization increasingly appears to control longevity and transitions (Vogel et al., 2022). The approach is physically grounded as $w$ is diagnosed from continuity and $w_e$ from the boundary-layer mass budget using only MISR inputs. This eliminates the need to use reanalysis model provided vertical air motion or other ancillary thermodynamic profiles. As a result, systematic and random uncertainties can be traced directly to the well-characterized MISR uncertainties in heights and winds (Horváth, 2013; Mueller et al., 2017; Mitra et al.,

2021; Loveridge and Di Girolamo, 2025). To our knowledge, this study represents the first direct spaceborne retrieval of cloud-top entrainment rates based solely on stereo-derived winds and heights.

The technique was applied to a stratocumulus deck off the United States West Coast. The MISR retrievals and ERA5 yielded similar qualitative picture of on average a weak descent and predominantly positive entrainment rates. Despite the differences in the temporal and spatial resolutions, MISR-retrieved $w$ and $w_e$ compare well against those from the ERA5. MISR retains

mesoscale structure in $H_T$ and winds that the comparatively smoother reanalysis model field cannot. Hence, divergence – and thus, $w$ values differ in detail even when the scene mean values are similar. Known anisotropy of the MISR along-track component also projects into the divergence (Horváth, 2013; Lonitz and Horváth, 2011).

Previous satellite retrievals of cloud-top entrainment typically utilized cloud-top height estimates from infrared sensors but estimates of vertical motion in those approaches were typically taken from a reanalysis into a mass budget consideration (e.g.,

Painemal and Zuidema, 2011; Zhu et al., 2025 a, b). Despite the overall coverage and benefits of such retrievals, the most influential term, i.e., vertical motion ($w$) is not an observational estimate. Rather, it is taken from a reanalysis model, so the largest and least traceable uncertainty in the retrieval method sits outside the observations. The proposed retrieval approach differs from those efforts by keeping the calculation strictly grounded within the observations. Here, both cloud top height and wind estimates come from the same stereo geometry, and both carry tight validation. For low clouds, MISR heights are accurate



to a few hundred metres and winds to a few metres per second, with biases that are small and stable (Horváth, 2013; Mitra et al., 2021). Those properties allow us to propagate uncertainties through to $w$ and $w_e$ at each grid point. In the case examined, typical random uncertainties are about 0.7 cm s$^{-1}$ for $w$ and 0.5 cm s$^{-1}$ for $w_e$; even screening for fractional uncertainties less than 25% retains upwards of 85% of all retrievals. These features make the retrieval suitable for both case studies and climatological aggregation (Vallejos et al., 2014).

Although the technique is theoretically grounded, the retrieved entrainment rates were negative for 29% of samples for the case. These samples mostly corresponded to instances of positive upward motion. During these instances, for the entrainment rates to be positive, the cloud top height advection terms in equation (6) needs to be greater than the vertical air motion. As the advection terms was largely negligible, this resulting balance was between the mesoscale subsidence and the entrainment rates. For the case presented herein, the advection terms (gradients) for cloud top height were calculated at 0.4°. Perhaps during the

instances of mesoscale ascent, a different window is required for calculating cloud top advection terms, or observations of CTH at spatial resolution finer than 17 km could be needed. This would be an additional but worthwhile effort when the retrievals will be used for conducting process-level studies.

Cloud-motion vectors are an underused bridge between cloud physics and dynamics. Recent work has shown that the MISR CMV record can reveal circulation changes on decadal scales, underscoring the benefits of analysing these vectors at their

native resolution and with care for their error budgets (Di Girolamo et al., 2025). The present study takes a complementary step: using those same CMVs at low cloud-top to observe entrainment directly, with uncertainties that are explicit and portable. An operational product based on this method is straightforward to envision. For each MISR overpass, inputs would be interpolated to a 0.2° grid to deliver gradients, divergence, the height-advection term, and then derive $w$ and $w_e$. Each grid cell would carry propagated random and systematic uncertainties and quality flags (e.g., a fractional-uncertainty mask, neighbour

counts for gradients, thin-cirrus indicators). Monthly aggregates on a coarser grid could then be used to generate climatology. However, caution needs to be taken while interpreting such climatology and using them for model evaluation as Terra observes at a fixed local time (~10:30 LT), so the product will not resolve the diurnal cycle.

The scientific use cases of such a product can be varied. Process modellers can compare instantaneous retrievals of $w$ and $w_e$ with large-eddy simulation (LES) or single-column model output for the same scenes to test entrainment parameterizations

and mixing assumptions (Bretherton et al., 2007; Mellado, 2017; Wood, 2012). Climatologists can build a 2000–2023 record of $w$ and $w_e$ for low clouds by region and season and examine how entrainment covaries with stability, surface forcing, and aerosol sensitivity (Nam et al., 2012; Vial et al., 2013; Grosvenor et al., 2018; Xu et al., 2022; Luo et al., 2020). As the Terra orbit has been very stable over this period, trends in cloud top heights, winds, subsidence, and entrainment could also be calculated. Such a trend analysis will shed insights on the coupling between low level cloudiness, mesoscale circulation and

entrainment, which is challenging for Earth System Models (Vial et al. 2016; Brient et al. 2016). The overall record could also be used to conduct in-depth studies by fusing it with complementary data from heavily instrumented sites such as the





Atmospheric Radiation Measurement (ARM) observatories or field campaigns. High resolution observations from Doppler radars, lidars, and surface instruments can provide independent checks on retrieved vertical air motion and entrainment rates. For example, the ARM Eastern North Atlantic (ENA) site located in the Azores routinely experiences marine stratocumulus
clouds and hence is an ideal site for performing such evaluation.

Overall, this case study is encouraging. It shows that a stereo-only retrievals can deliver cloud-top entrainment with uncertainties small enough to be informative, and with mesoscale detail that matters for stratocumulus. It provides an observational counterpart to other estimates of these variables that are independent of model vertical motion and that comes with traceable errors. Turning this into a regular product would make a long, homogeneous record of $w$ and $w_e$ available to the
community. Such a record would help stress-test entrainment schemes, clarify regional and seasonal variability, and connect low-cloud physics to the circulation changes now being detected from CMVs themselves (Di Girolamo et al., 2025).

**Appendix**

The retrieval technique (Section 2.2.1) makes few assumptions, and we explore their validity here. The mesoscale vertical air motion is derived from the continuity equation applied to the MISR CMV reported wind vectors (Eq. 4). The derivation
assumes that the gradients in the horizontal winds at the cloud top are the same throughout the PBL, and the mesoscale vertical air motion is zero at the surface. The profiles of gradient in horizontal winds $(\frac{\partial u}{\partial x}, \frac{\partial v}{\partial y})$ derived from the ERA5 data for the case (Figure A1a) shows that the values are uniform throughout much of the boundary layer (expressed in normalized units $\xi = {}^{z}/_{PBL\ Height}$, where $z$ is height above the surface). The ERA5 $w$ profile (derived from ERA5 reported pressure tendency through hydrostatic conversion) varies only weakly with height over the same extent of the ERA5 reported PBL depth. This
region is dominated by gentle subsidence with mean vertical air motion at the surface and CTH differing only by a small amount (surface: -0.05$\pm$0.54 cm s$^{-1}$; CTH: -0.35$\pm$0.74 cm s$^{-1}$).



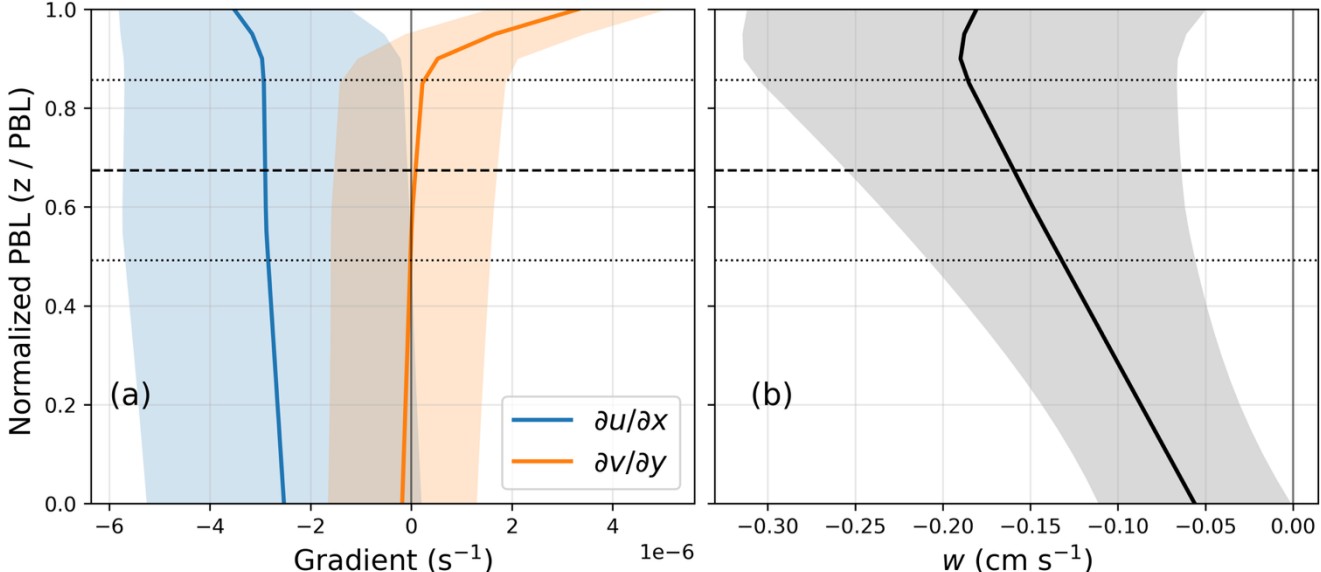

**Figure A1:** ERA5 mean profiles on a normalized PBL coordinate (z/BLH) (a) Horizontal wind-gradient components $\frac{\partial u}{\partial x}$ (blue) and $\frac{\partial v}{\partial y}$ (orange); (b) vertical velocity $w$ (black) (positive upward). Solid curves show the mean and shaded envelopes denote one standard deviation variability. Horizontal dashed lines show mean (thick) and standard deviation (thin) of MISR retrieved heights (transformed to the normalized PBL coordinate).

As noted in Fig. 4, MISR CTH are lower than ERA5 PBL depths on average by $189 \pm 87$ m, over the region considered. Thus, MISR CTHs are found to lie on average at $\xi = 0.68 \pm 0.18$. Interestingly, Figures A1a and A1b show that MISR-derived stereo CTHs are a better estimate than ERA5 reported PBL depths (i.e., $\xi = 1$) as the typical structure of a well-mixed boundary layer ($\frac{\partial u}{\partial x}, \frac{\partial v}{\partial y}$ uniform with height and a nearly linear profile of $w$) is found below MISR CTHs. In Fig. A2, we compare two independent ERA5 vertical air motion estimates at PBL top:

(i)     a continuity estimate, $w_c = -H_T \left( \frac{\partial u}{\partial x} + \frac{\partial v}{\partial y} \right)$ from ERA5 winds and PBL heights using Eq. 4

(ii)    a hydrostatic estimate, $w_h$, derived from ERA5 reported pressure tendencies.





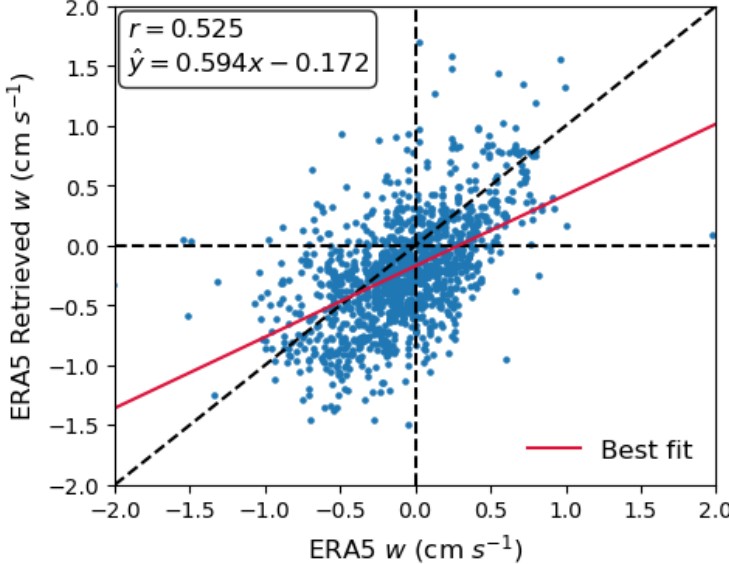


**Figure A2:** Scatter comparison of ERA5 vertical velocity at PBL top against the continuity-based ERA5 retrieval (from horizontal wind divergence). Each marker is a co-located grid cell. The dashed black line is the 1:1 line; the solid red line is the least-squares fit (equation shown). The upper-left inset reports the Pearson correlation coefficient (r).

The average difference between the two estimates was -0.15±0.4 cm s$^{-1}$. There is a moderate correlation between the two estimates ($r = 0.5$) with the best-fit line showing a small offset of 0.18 cm s$^{-1}$. If the assumptions involved were completely self-consistent, we would see an exact 1:1 line. These deviations from that ideal scenario are likely due to implementation artefacts (e.g., finite-difference gradients on the native grid) and assumptions in deriving $w_h$ (e.g., hydrostatic balance and ideal gas law with virtual temperature). Crucially however, the sign agreement between $w_c$ and $w_h$ and their near-linear co-580 variation show that the continuity estimate captures the same mesoscale signal at PBL top as the hydrostatic form to first order.

**Author Contribution:**

AM and VG envisioned the retrieval technique. AM downloaded the data and performed the analysis. AM and VG equally 585 contributed to the writing of the manuscript.

**Data Availability:**

MISR CMV data are publicly available at the NASA Langley Atmospheric Science Data Center (https://asdc.larc.nasa.gov/project/MISR/MI3MCMVN_2). ERA5 Reanalyzes are publicly available through the European 590 Center for Medium-Range Weather Forecast (ECMWF) Climate Data Store (CDS) website (https://cds.climate.copernicus.eu/cdsapp#!/home). SATCORPS GEO Edition 4 GOES NH data was accessed through searching on NASA's Earthdata website (https://search.earthdata.nasa.gov/search?q=SATCORPS%20GOES%20NH). All



MODIS data are publicly available through the Level 1 and Atmosphere Archive and Distribution System of NASA Goddard Space Flight Center (https://ladsweb.modaps.eosdis.nasa.gov/archive/allData/61/).


**Code Availability:**

All code is developed using open-source libraries in Python and will be made available upon request.

**Competing Interest:**

The authors declare that they have no conflict of interest.

**Acknowledgements**:

We gratefully acknowledge the computing resources provided on Improv, a high-performance computing cluster operated by the Laboratory Computing Resource Center at Argonne National Laboratory.


**Financial Support:**

This research was supported by the Argonne National Laboratory (ANL)'s Laboratory Directed Research and Development (LDRD) program, and the U.S. Department of Energy's (DOE) Atmospheric System Research (ASR), an Office of Science, Office of Biological and Environmental Research (BER) program, under Contract DE-AC02-06CH11357 awarded to Argonne

National Laboratory.

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
