# Peer review of "Towards Retrieving Cloud Top Entrainment Velocities from MISR Cloud Motion Vectors"

_EGUsphere, 2025_

## Author Comment (AC1)

We thank the reviewer for giving detailed critiques of the manuscript. Addressing these comments have led to substantial improvements in the manuscript. Please find below the reviewer comments in **Bold**, our responses to them in regular text, and changes to text if any in *Italics*. Thank you.

**This is an interesting study and well-written manuscript that describes and tests a new physically based technique for retrieving vertical and entrainment velocities at the top of a stratocumulus layer over the ocean using cloud motion vectors. It is an achievement to estimate the vertical velocity and entrainment velocity "strictly grounded within the observations". The work is methodical, the results demonstrate the potential usefulness of the new approach, and the manuscript is well-structured. Entrainment velocity is an important parameter for understanding and modelling ABL processes, and the exchange of heat, moisture, and pollutants at the earth's surface as well as at the top of the ABL. The manuscript presents a valuable work that is both timely and important, and will be a significant contribution to the field.**
Thank you!

**Major limitation:**
**One significant drawback of the current study is the use of ERA5 as a reference for the satellite retrievals. ABL, low-level clouds, and top entrainment are examples of small-scale processes that are poorly resolved in the reanalysis and depend on highly uncertain model physics (parameterization schemes). It is not very compelling to evaluate direct estimations of CTH, w, and $w_e$ based on satellite data using reanalysis as a reference.**
**Furthermore, the rationale for data selection is somewhat illogical since it stresses that the new approach eliminates the need for reanalysis to provide vertical air velocity. This is a noteworthy development, however, it is unclear why ERA5 may then be used as a reference.**
We thank the reviewer for this comment, similar comment was made by the other reviewer. We fully agree that as ERA5 is a numerical reanalysis model, the PBL, w and w_e from it cannot be treated as observational truth. In this work, ERA5 is not used to validate the MISR-based retrievals of cloud-top vertical velocity (w) or entrainment velocity (w_e), nor do we claim that agreement with ERA5 implies absolute accuracy. Rather, ERA5 is used strictly as a sanity check and as an independent physical reference for the satellite derived estimates. Hence, the title of the article is "Towards retrieving cloud top entrainment velocities...", rather than "Retrievals of entrainment velocities ...". Perhaps due to us being unclear, these points did not come across clearly in the submitted version of the manuscript. In the revised version we have clarified this issue by adding the following sentence to the introduction and summary sections.

*Introduction Section:*
*"The present study does not attempt a formal validation of the retrieved w or w_e. Comparisons with ERA5 reanalysis model are solely made for physical consistency check and not as an observational benchmark. The uncertainty estimates reported herein arise*

*from analytical propagation of known MISR cloud-top height and wind uncertainties, which have been independently characterized in prior studies (e.g., Mueller et al., 2017; Mitra et al., 2021). The goal of this work is therefore to demonstrate the feasibility and internal consistency of a stereo-only observational retrieval of entrainment, rather than to establish absolute accuracy."*

*Summary Section:*
*"While this study demonstrates the feasibility of retrieving cloud-top vertical velocity and entrainment rates from MISR observations alone, it does not constitute a formal validation of the retrieved magnitudes. The comparison with ERA5 serves only as a physical consistency check and contextual reference. The ERA5 reported PBL height derived from the potential temperature gradients are used herein, while those derived from relative humidity (RH) gradient might be more applicable for low-level clouds (Von Engeln and Teixeira, 2013). A rigorous evaluation would require independent observations of vertical motion, inversion height, and entrainment at comparable spatial scales. Long-term ground-based facilities such as the ARM Eastern North Atlantic (ENA) site provide a promising opportunity in this regard, combining frequent radiosondes with cloud radar and lidar observations in marine stratocumulus regimes. A future evaluation strategy could involve regime-based compositing of MISR overpasses collocated with ENA observations and statistical comparison of estimates of cloud top heights, vertical air motion, and entrainment rates derived from ground-based instruments. Such an approach would respect inherent spatial and temporal mismatches while providing an independent test of the MISR retrieval framework."*

**Minor comments:**
**Line 12-13: "output from European Center for Medium-range Weather Forecasting (ECMWF) reanalysis model (ERA5)" -> "European** Centre **for Medium-range Weather Forecasting (ECMWF) reanalysis (ERA5)"**
Thank you for catching this, we have fixed the typo.

**ERA5 provides reanalysis fields derived from a combination of observations and modelling, and as such cannot be considered as model output. The model used in a chain of processes to produce the reanalysis is IFS.**
Agreed, the sentence has now been rephrased as "... data from the European Centre for Medium-range Weather Forecasting (ECMWF) reanalysis (ERA5)."

**Line 19: „a generate a global climatology"-> „generate a global climatology"**
Thank you. Corrected.

**Page 13 – It should be mentioned here already that ERA5 estimates of cloud top height are higher and less spatially variable than MISR CTH.**
We appreciate the suggestion. It has been implemented.

**Line 42: Explanation of what a Twin Otter is and some reference to its use in meteorological observation should be added. The same is valid for UAS.**

The Center for Interdisciplinary Remotely Piloted Aircraft Studies (CIRPAS) at the Naval Postgraduate School (NPS) in Monterey, CA owns and operates an instrumented Twin Otter aircraft that has been long used to study low-cloud processes. It is an unpressurized low-flying and slow moving (~50 m/s) aircraft that can make in situ measurements of aerosols, cloud, thermodynamic, dynamic, and radiative properties. Data from multiple Twin Otter field campaigns have been summarized in Sorooshian et al. (2018). We particularly highlight two studies that have explicitly derived entrainment rates, Gerber et al. (2013) for the Physics of Stratocumulus Top (POST) field campaign, and Norgren et al. (2016) for the Gulf of Mexico Atmospheric Composition and Climate Study (GoMACCS) field campaign.

The Unmanned Aerial Systems (UAS) can make observations of thermodynamic and at times that of cloud properties. UAS can fly in low to moderate wind conditions at speeds ranging from 10-36 m/s. We want to highlight observations collected during the recent TRacking Aerosol Convection interactions ExpeRiment (TRACER) field campaign summarized in Lappin et al. 2025.

Sorooshian A, MacDonald AB, Dadashazar H, Bates KH, Coggon MM, Craven JS, Crosbie E, Hersey SP, Hodas N, Lin JJ, Negrón Marty A, Maudlin LC, Metcalf AR, Murphy SM, Padró LT, Prabhakar G, Rissman TA, Shingler T, Varutbangkul V, Wang Z, Woods RK, Chuang PY, Nenes A, Jonsson HH, Flagan RC, Seinfeld JH. A multi-year data set on aerosol-cloud-precipitation-meteorology interactions for marine stratocumulus clouds. Sci Data. 2018 Feb 27;5:180026. doi: 10.1038/sdata.2018.26. PMID: 29485627; PMCID: PMC5827690.

Gerber, H., G. Frick, S. P. Malinowski, H. Jonsson, D. Khelif, and S. K. Krueger (2013), Entrainment rates and microphysics in POST stratocumulus, J. Geophys. Res. Atmos., 118, 12,094–12,109, doi:10.1002/jgrd.50878.

Norgren, M. S., Small, J. D., Jonsson, H. H., and Chuang, P. Y.: Observational estimates of detrainment and entrainment in non-precipitating shallow cumulus, Atmos. Chem. Phys., 16, 21–33, https://doi.org/10.5194/acp-16-21-2016, 2016.

Lappin, F., de Boer, G., Klein, P., Hamilton, J., Spencer, M., Calmer, R., Segales, A. R., Rhodes, M., Bell, T. M., Buchli, J., Britt, K., Asher, E., Medina, I., Butterworth, B., Otterstatter, L., Ritsch, M., Puxley, B., Miller, A., Jordan, A., Gomez-Faulk, C., Smith, E., Borenstein, S., Thornberry, T., Argrow, B., and Pillar-Little, E.: Data collected using small uncrewed aircraft systems during the TRacking Aerosol Convection interactions ExpeRiment (TRACER), Earth Syst. Sci. Data, 16, 2525–2541, https://doi.org/10.5194/essd-16-2525-2024, 2024.

**Line 55: "of of" -> "of"**
Thank you for the catch. Corrected.

**Line 74: "the retrieval method can be extended to the global: - Can it also be applied over land?**
YES, but with caveats. For low clouds over land, especially over bright surfaces (snow, ice, desert) or for very low clouds, MISR stereo may misidentify surface features as clouds. This can however be flagged and screened using a high-resolution and modern DEM.

**Line 207: Please remove "model" here, and also remove "model output" from Table 1.**
Accepted, the words have been removed.

---

## Author Comment (AC2)

We thank the reviewer for giving detailed critiques of the manuscript. Addressing these comments have led to substantial improvements in the manuscript. Please find below the comments in **Bold**, our responses to them in regular text, and changes to text if any in *Italics*. Thank you.

**The manuscript describes a novel method for calculating entrainment rates (W_e) in marine stratocumulus cloud regimes, with the use of cloud top height and winds estimates from MISR. The methodology is sound, and the new W_e has the advantage of not relying on numerical weather predictions of wind speed and large-scale velocity. Moreover, MISR winds allow for the estimation of vertical velocity. The method is unique and offers a valuable independent estimation. In general, it is an interesting and well written paper.**

Thank you!

**Specific comments**

**My general criticism is on the way uncertainties are estimated from observations and the use of ERA-5 reanalysis to lend confidence to the MISR W_e. Comparisons against ERA-5 are problematic because a numerical model cannot validate observations. Indeed, if ERA-5 outputs were accurate enough, then independent satellite observations would be unnecessary. At the very best, the comparison with ERA-5 is a sanity check, which should not be used to make inferences about accuracy of different satellite datasets. Another aspect to consider is how the PBL height from ERA-5 is estimated, which is based on the bulk Richardson number method. Von Engeln and Teixeira (2013) show that this PBL height underestimates the inversion height derived from the relative humidity (RH) profile by more than 200 m in the subtropics. Instead, they conclude that the bulk Richardson method for deriving PBL height might be more representative of the cloud base height. Considering a typical cloud depth of around 200 m- 400 m, I would estimate a low bias in ERA-5 PBL height to be around the same range. The PBL height issue could be remediated by estimating the inversion height applying the method(s) described in Von Engeln and Teixeira (2013). Possibly, the new PBL height would also impact the selection of the vertical level for extracting the vertical velocity from the model. Even if these corrections are implemented, the central issue of validating MISR retrievals with reanalysis model outputs remains. Ideally, one would validate the data with lidar/radar observations and/or radiosondes. But I am not sure if these datasets actually exist for the period of study in the manuscript. Without a more rigorous validation dataset, the authors should revise the manuscript to state that the results are reasonable, but I don't think enough evidence is provided to: a) validate the magnitude of W_e, and b) to determine that MISR W_e is better than methods derived from the mixed-layer budget equation and retrievals from MODIS/GOES.**

We thank the reviewer for this comment. We fully agree that as ERA5 is a numerical reanalysis model, the PBL, w and w_e from it cannot be treated as observational truth. In this work, ERA5 is not used to validate the MISR-based retrievals of cloud-top vertical velocity

(w) or entrainment velocity (w_e), nor do we claim that agreement with ERA5 implies absolute accuracy. Rather, ERA5 is used strictly as a sanity check and as an independent physical reference for the satellite derived estimates. Hence, the title of the article is "Towards retrieving cloud top entrainment velocities...", rather than "Retrievals of entrainment velocities ...". Perhaps due to us being unclear, these points did not come across clearly in the submitted version of the manuscript. In the revised version we have clarified this issue by adding the following sentence to the introduction and summary sections.

*Introduction Section:*
*"The present study does not attempt a formal validation of the retrieved w or w_e. Comparisons with ERA5 reanalysis model are solely made for physical consistency check and not as an observational benchmark. The uncertainty estimates reported herein arise from analytical propagation of known MISR cloud-top height and wind uncertainties, which have been independently characterized in prior studies (e.g., Mueller et al., 2017; Mitra et al., 2021). The goal of this work is therefore to demonstrate the feasibility and internal consistency of a stereo-only observational retrieval of entrainment, rather than to establish absolute accuracy."*

*Summary Section:*
*"While this study demonstrates the feasibility of retrieving cloud-top vertical velocity and entrainment rates from MISR observations alone, it does not constitute a formal validation of the retrieved magnitudes. The comparison with ERA5 serves only as a physical consistency check and contextual reference. The ERA5 reported PBL height derived from the potential temperature gradients are used herein, while those derived from relative humidity (RH) gradient might be more applicable for low-level clouds (Von Engeln and Teixeira, 2013). A rigorous evaluation would require independent observations of vertical motion, inversion height, and entrainment at comparable spatial scales. Long-term ground-based facilities such as the ARM Eastern North Atlantic (ENA) site provide a promising opportunity in this regard, combining frequent radiosondes with cloud radar and lidar observations in marine stratocumulus regimes. A future evaluation strategy could involve regime-based compositing of MISR overpasses collocated with ENA observations and statistical comparison of estimates of cloud top heights, vertical air motion, and entrainment rates derived from ground-based instruments. Such an approach would respect inherent spatial and temporal mismatches while providing an independent test of the MISR retrieval framework."*

**MISR vs GOES/MODIS: I fully agree with the authors about the crucial differences between MISR and satellite infrared sensors, and the advantages of the MISR products (independent of weather models and directly estimated from geometric considerations). Having said that, the manuscript needs to provide more compelling evidence that MISR retrievals of cloud height in the boundary layer show real improvements relative to MODIS/GOES in terms of bias and/or root mean square errors. As described in comment #1, the use of ERA-5 PBL height is not ideal for validating**

**satellite-based height. Also, table 1 and Mitra et al. (2021) show that for clouds with tops < 5km, MISR is biased low by 240 m, whereas the MODIS bias is + 60 m. Along the same lines, an analysis we conducted in LaRC using GOES-15 retrievals demonstrated that GOES-15 cloud top height are nearly unbiased relative to radar data collected over the Northeast Pacific Ocean (RMSE< 200m, Painemal et al. 2017).**

The MISR and MODIS cloud top height retrievals were compared in Mitra et al. (2021). Broadly the comparisons revealed:

1) MISR has a consistent bias of -240 m, but the sign of the bias itself is not a strong function of the optical depth of the low cloud. Whereas, while MODIS (and potentially other IR sensors) ostensibly has a 'lower' bias for low clouds, that is an average effect of larger biases of around $\pm500$ m, with generally a positive bias for optically thicker and negative bias for optically thinner clouds. This is probably facilitated through an impact of the optical depth of the low cloud in modulating the cloud-top temperature inversion, which determines the IR cloud-top and 2) MISR low-cloud bias is not affected by the presence or absence thereof high thin clouds (i.e., cirrus), whereas IR retrievals are affected by them.

Thus, although the mean error in MISR cloud top heights appears bigger in absolute terms (i.e., more negative), it is self-consistent and unperturbed by other elements in the scene as compared to estimates from IR techniques that require ancillary inputs with their own uncertainties. As a result, the error-budget closure achieved for the MISR low CTH is directly responsible for the error-budget closure of our entrainment retrieval.

The good agreement between the GOES-15 and radar derived cloud top heights during the MAGIC field campaign as reported by Painemal et al. 2017 is not surprising given the optically thick stratus and very few Cirrus clouds sampled during the campaign. Zhou et al. (2015 J. Clim) reported the Cirrus cloud cover to be less than 10% and the average low cloud thickness of 230 m.

We agree with the reviewer that a comparison of GOES, and MISR derived cloud top heights, and associated entrainment rates with those from ground-based cloud radars for variety of low cloud optical depths and high-level cloudiness is warranted. However, such a comparison is outside the scope of the current work. Hence, we have added the following sentence to the summary section.

*While this study demonstrates the feasibility of retrieving cloud-top vertical velocity and entrainment rates from MISR observations alone, it does not constitute a formal validation of the retrieved magnitudes. The comparison with ERA5 serves only as a physical consistency check and contextual reference. A rigorous evaluation would require independent observations of vertical motion, inversion height, and entrainment at comparable spatial scales. Long-term ground-based facilities such as the ARM Eastern North Atlantic (ENA) site provide a promising opportunity in this regard, combining frequent radiosondes with cloud radar and lidar observations in marine stratocumulus regimes. A future evaluation strategy could involve regime-based compositing of MISR overpasses collocated with ENA observations and statistical comparison of estimates of cloud top heights, vertical air*

*motion, and entrainment rates derived from ground-based instruments. Such an approach would respect inherent spatial and temporal mismatches while providing an independent test of the MISR retrieval framework.*

However, we have now compared the GOES derived entrainment rates per Painemal et al. (2017) to those from MISR (Figure 1 below).

[Figure]

**Figure 1**: (a) GOES $w_e$ retrievals (b) ERA5 $w_e$ (c) Differences between GOES and ERA5 $w_e$ (d) GOES CTH advective term (e) ERA5 PBL height advective term (f) Differences between GOES and ERA5 PBL advective terms. In these panels, GOES retrievals are done using GOES estimates of CTH (using GOES cloud-top temperature and ERA5 SST) and ERA5 u and v vectors. Cloud top heights above 1.5 km (near the coast) are deemed contaminated by cirrus and removed.

The results of the comparison between the GOES+ERA5 retrievals and our MISR-based retrievals have now been summarized in Section 3.2, as follows.

*As a further sanity check of the retrieved values, we compared MISR-derived entrainment rates against independent estimates obtained using GOES cloud-top heights and winds following the mass-budget framework of Painemal et al. (2017). In this approach, GOES–SST cloud-top heights (Fig. 4) are combined with ERA5 horizontal wind fields sampled at the GOES–SST cloud-top heights to estimate cloud-top vertical velocity and entrainment rates using the same continuity and mass-budget formulation described in Sect. 2.2.1. Over the analysis domain, GOES-derived entrainment velocities ($w_e^{GOES}$) exhibit the same sign and mesoscale spatial coherence as the MISR-derived values, indicating consistent diagnosis of entraining versus detraining regions across the cloud deck. However, the domain-mean $w_e^{GOES}$ (0.44$\pm$0.35 cm s$^{-1}$), is larger than the corresponding MISR ($w_e^{MISR}$) mean, yielding a mean difference of mean difference ($w_e^{MISR} - w_e^{GOES}$) of -0.28$\pm$0.35 cm s$^{-1}$. This offset is not unexpected, given the reliance of the GOES-based estimate on infrared cloud-top height retrievals and reanalysis winds, both of which tend to smooth cloud-top gradients and can project height-assignment differences into the diagnosed divergence and vertical motion. In contrast, the height-advection term (A) derived from GOES remains weak, with a domain mean of 0.05 ± 0.38 cm s$^{-1}$, similar to the MISR-based estimate and close to zero across most of the scene. Thus, while the magnitude of $w_e^{GOES}$ differs from $w_e^{MISR}$, the qualitative behavior and spatial organization of entrainment are robust across independent satellite frameworks.*

**All in all, this is a nice paper with an interesting method for deriving entrainment rates from MISR. My suggestion to the authors is to revise the manuscript and clarify that no rigorous validation of the retrievals are provided with independent datasets (instead of models) and therefore, uncertainties cannot be estimated with the necessary detail.**
Thank you for the kind words and we have now revised the manuscript text and abstract to reflect this. In addition, to the paragraph to the summary section mentioned above, we have added the sentence below to the abstract.

*The uncertainties in the utilized CTHs and CMVs are propagated to derive systematic and random retrieval uncertainties in vertical air motion and entrainment rates. Comparison of the retrieved vertical air motion and entrainment rates with estimates from an independent dataset during variety of weather states is warranted for fully validating the retrieval technique.*

**Other publications: I was a bit surprised that entrainment rates from other methods were not discussed and compared with the MISR rates (Cadwell et al., Faloona et al., Ghate et al, Wood and Bretherton from the reference section). For instance, all these studies derive W_e< 1 cm/s; is this consistent with the results from MISR? Also, Cadwell et al. resolve the diurnal cycle, and therefore, Terra overpass time can be matched with their diurnal cycle figure. Moreover, Painemal et al. (2017) resolve the diurnal cycle over the same region of study and computed spatial maps for different hours. In sum, there is enough available references that should be discussed in the context of estimating and validating entrainment rates. Cadwell et al. and Painemal et al. 2017 that entrainment rates can be, at times, slightly negatives (it is not clear to me whether this**

**is issue associated with the mixed-layer budget equation). Lastly, the entrainment rates in Painemal et al. (2017) were validated over the Western North Atlantic by Tornow et al. (2022) with the use of airborne observations.**

This is a very good suggestion, especially as we have not performed independent evaluation of the MISR entrainment rates. We now have added the sentence below to compare the derived entrainment rates to those from previous works. We have also compared the retrieved mesoscale vertical air motion to the past estimates of Bony and Stevens (2019). Thank you.

*The domain mean and standard deviation of mesoscale vertical air motion was -0.36±0.65 cm s$^{-1}$, which is much weaker than the 50 hPa hour$^{1}$ (11.7 cm s$^{-1}$) reported by Bony and Stevens (2019) in the tropics. The domain mean and standard deviation of entrainment velocity was 0.28±0.40 cm s$^{-1}$. This estimate is in the ball-park of the estimates reported by Painemal et al. (2017) of -0.4 to 0.8 cm s-1 in the South Atlantic, 0.723 cm s$^{-1}$ reported by Ghate et al. (2019) in the North Pacific, 0.57 cm s-1 by Faloona et al. (2004) in the coastal California, 0-0.5 cm s$^{-1}$ as reported by Caldwell et al. (2005) in the South Pacific. As expected, the MISR entrainment velocity estimates for the stratocumulus case analyzed herein are much weaker than the 0-20 cm s$^{-1}$ entrainment velocities reported by Tornow et al. (2022) for a cold-air outbreak case. The MISR estimates were also within the range of 0-1 cm s$^{-1}$ entrainment velocities reported by Albrecht et al. (2016) for a continental stratocumulus case. It should be noted that due to MISR's overpass at 10:30 am, the derived entrainment rates are likely at the lower end given the strong diurnal cycles reported by Painemal et al. (2017) and Caldwell et al. (2005). The GOES derived entrainment rates by Painemal et al. (2017) for the southeast Atlantic Stratocumulus deck were negative between 07-13 Local Time, which partly explains the negative values of MISR entrainment rates as they are made at 10:30 LT.*

**Other comments:**
**It would be easier to extract quantitative information from the figures if the authors adopt a color scale/palette with discrete colors (e.g. 12 or 14 colors).**
Thank you for the suggestion. A discretized color scale will be applied in the final figure uploads and in the revised manuscript.

**Line 35, Grosvenor et al. does not explicitly analyze the effect of entrainment.**
While this is true, the paper does provide a comprehensive review of the state of cloud microphysical studies and does highlight the role of entrainment in modulating subadiabaticity (Section 2.3.3 of the paper), which is important in the context.

**Line 50, the citation does not exist.**
Thank you for highlighting this. The erroneous citation is now removed.

**Line 55 Minnis et al does not discuss entrainment rates.**

The Minnis et al. citation was meant to highlight the underlying cloud-top estimates and their uncertainties. To clarify this point, the citation has been now moved earlier in the sentence. Thank you.

**Line 58: you mean "offers a unique dataset"?**
Exactly, thank you. That is certainly a better choice of word and has been updated.

**Line 96: you mean "The above equation is integrated…"**
Thank you. This has been updated.

**Line 127 and equation (11). Could you be more explicit about the way equation (11) is derived?**
This is clarified now with the slightly extended explanation as

*Meanwhile, to calculate the precision in the advection term A, we take the derivative of the mathematical form of A with every term in its right-hand side in Eq. 8. to derive*

$$\frac{\partial A}{\partial u} = \frac{\partial H_T}{\partial x}; \quad \frac{\partial A}{\partial \left(\frac{\partial H_T}{\partial x}\right)} = u; \quad \frac{\partial A}{\partial v} = \frac{\partial H_T}{\partial y}; \quad \frac{\partial A}{\partial \left(\frac{\partial H_T}{\partial y}\right)} = v \tag{11}$$

**Line 150. Why is the standard deviation a measure of error?**
Standard deviation measures error by quantifying the typical spread or dispersion of individual data points around the mean. Thus, it is crucial in quantifying the error introduced in any set of observations due to uncertainties created by random fluctuations about an average behavior. For example, slight fluctuations in cloud optical depth will influence how deep into the 'true' cloud-top the satellite will peer and this introduces a random variation around the mean MISR bias of -240 m. We follow here the definition of the standard error of the mean of a distribution that is the ratio of standard deviation to the square root of the number of samples used.

**Line 158, what is sampling uncertainty?**
Sampling uncertainty is the difference between a sample's statistic and its population's true parameter, caused by random errors during the sampling process. It arises because a sample is only a subset of the total population and may not perfectly represent it, especially if the population is heterogeneous. This uncertainty is quantified using methods like the standard error and is reduced by increasing the sample size.

**Line 171 and eq (22). "A" is already used for advection. Please, use a different symbol for denoting area.**
Thank you for pointing this out. We now use 'AR' instead of 'A' for area to prevent confusion with advection.

**Line 178, you mean eq. (22)?**
Thank you for highlighting this. We actually meant to refer to both Equations 21 and 22. This has now been updated.

**Table 1: For SatCORPS GOES, the satellite is GOES-15, and the pixel resolution is 4km at nadir. The 8km resolution refers to a subsampling (every other pixel) applied to the map, but the pixel resolution is 4km. Also, the nominal uncertainty of 500 m does not seem correct for boundary layer clouds (cloud tops < 3km).**
Thank you for pointing this out. We now mention in the text that the actual pixel resolution is 4km, while the data are subsampled at 8km resolution. The GOES reports cloud top temperatures from which the cloud top heights are calculated. The value of 500m was from the Table 6 in theoretical basis document listed below. The Table reports errors for low-level clouds with emissivity greater than 0.8. Thank you.

https://www.star.nesdis.noaa.gov/goesr/documents/ATBDs/Baseline/ATBD_GOES-R_Cloud_Height_v3.0_Jul2012.pdf

**Page 11. A common way of removing noise in the geophysical fields is smoothing the variables using digital filters before estimating spatial gradients. Is spatial noise a relevant issue in the calculation of advection and divergence?**
This is a good point, however for the case-study reported here we did not have to do any additional smoothing/filtering apart from the re-gridding of the MISR data to a regular latitude-longitude grid with resolution of 0.2 degree. However, some spatial smoothing might need to be performed if the algorithm is implemented to develop long-term climatology. We have mentioned this in the summary section.

**Line 273 "However negative CTH will need to be converted to heights over the geoid for retrieval calculations in further iterations of this technique." How about pixels with cloud tops below 250 m? (it seems implausible that the cloud tops could be lower than 250 m). Does it mean that MISR CTH are always biased low? A 200 m underestimation could impact estimates from equation (4).**
Yes, MISR CTH are always biased lower than 'true' cloud top. The MISR CTH are defined with respect to mean sea level, not necessarily the surface level (hence, negative CTHs also possible). Please see Mitra et al. 2021 for further details.

**Line 299-301. I agree, infrared-based cloud top heights are biased under the presence of cirrus. However, this effect should be modest over the NE Pacific, especially if pixels with cloud heights > 3km or temp < 0˚C are removed from the analysis.**
While in principle this seems like a reliable way to mask cirrus clouds, practical analysis of scenes with thin cirrus over thicker low clouds may be more difficult. In many such instances (such as near the coast in our study scene) even though the final cloud-top height or temperature will fall within a regime consistent with low clouds (heights < 3km or temp > 0C), the presence of a thin cirrus overhanging the low cloud can still result in the overestimation of cloud-top heights or underestimation of cloud-top temperatures because of an effective

retrieval in the infrared. As a result, in our scene too, we note some cloud heights in the 2-5 km range in MODIS and GOES retrievals. However, MODIS retrievals are largely below 1 km or above 10 km, consistent both with the PBL heights in the region and the presence of a stray cirrus. In its own way, this contamination highlights a serious challenge of scaling IR techniques over many scenes and a relative strength of using the MISR dataset.

**Lines 310-313: I don't disagree that the MISR sampling of about 17 km is within the typical cloud object size in open/closed cells clouds. But I do not know if this really matters as it is unknown the spatial variability/scale of entrainment rates or vertical velocity.**
We agree with the reviewer that the spatial scales of entrainment rates and vertical velocities are not fully known. The sentence herein pertains to the closeness of the MISR resolution to the open/closed cells cloud sizes. Hence, we have left the sentence as it is.

**In light of comment # 1, the analysis in Figure 7 is not a validation of the MISR-based products. Since divergence is assumed constant with height, perhaps one could use ASCAT winds (9:30 LT morning pass) to compute divergence and compare it with its MISR counterpart.**
This is a very good suggestion. In addition to comparing the ASCAT reported winds and divergence to those from MISR, one can possibly derive the average boundary layer divergence by averaging the ASCAT and MISR reported divergences. However, the richness of variability of largescale vertical air motion as reported by Bony and Stevens (2019), might pose challenges to these efforts. We now have mentioned this in the summary section.

**Product vs retrieval: I have the impression that the MISR entrainment rate is a product, not a retrieval.**
Please accept our apologies but we don't fully understand the comment. The entrainment rates are derived from the MISR reported cloud top heights and the cloud motion vectors, so it fits the traditional definition of retrieval. The entrainment rates can be reported as a data product by applying the algorithm to all 22+ years of data, but that will be a significant effort.

**References**
**von Engeln, A., and J. Teixeira, 2013: A Planetary Boundary Layer Height Climatology Derived from ECMWF Reanalysis Data. J. Climate, 26, 6575–6590, https://doi.org/10.1175/JCLI-D-12-00385.1.**
**Painemal, D., K.-M. Xu, R. Palikonda, and P. Minnis (2017), Entrainment rate diurnal cycle in marine stratiform clouds estimated from geostationary satellite retrievals and a meteorological forecast model, Geophys. Res. Lett.,44, 7482–7489, doi:10.1002/2017GL074481.**
**Tornow, F., Ackerman, A. S., Fridlind, A. M., Cairns, B., Crosbie, E. C., Kirschler, S., et al. (2022), Dilution of boundary layer cloud condensation nucleus concentrations by free tropospheric entrainment during marine cold air outbreaks, Geophysical Research Letters, 49, e2022GL098444.**
We have added these references to the manuscript. Thank you.